# CrGSTA: Cross-domain Root causal Graph Spatial-Temporal Attention Network

## Abstract

Modern monitoring systems generate massive, high-dimensional time series where failures rarely remain isolated but cascade across interdependent components. Identifying their true origins requires more than anomaly detection; it requires interpretable models that disentangle causal structure from noisy signals. While Granger causality has gained traction for root cause analysis (RCA), existing neural methods often rely on multilayer perceptrons applied independently at each time step, which increases parameter counts, struggles with long-range dependencies, and overlooks seasonal and periodic patterns. We introduce CrGSTA (Cross-domain Root causal Graph Spatial-Temporal Attention Network), a scalable and interpretable framework that unifies time- and frequency-domain representations through cross-domain attention. CrGSTA employs graph-based spatio-temporal attention to capture directional dependencies, while frequency-aware features recover periodic structure. A lightweight self-attention decoder reconstructs dynamics, ensuring deviations are attributed to true root causes rather than propagated effects. We conduct experiments along three dimensions: temporal scalability, spatial scalability, and ablations on domain contributions and fusion strategies. On multiple synthetic and real-world datasets, CrGSTA new state of the art achieving up to 13% Avg@10 improvement by leveraging wider temporal windows with only 8.5M parameters compared to (200M+) of other baselines. By explicitly coupling temporal and frequency cues, CrGSTA balances accuracy, interpretability, and efficiency for RCA in complex monitoring environments, providing a foundation for more resilient and transparent analysis in real-world systems. https://github.com/crgsta2025/CrGSTA

## 1 Introduction

As digital infrastructures grow in scale and complexity, system failures are no longer isolated incidents but often trigger cascades of anomalies that spread across tightly coupled components Altenbernd et al. (2025). These anomalies, while infrequent, can severely disrupt application availability and compromise service reliability Nagalapatti et al. (2025). Traditional anomaly detection methods provide early warning signals, yet they fall short in answering the critical question of *why* the anomaly occurred Chen et al. (2019). Without this capability, operators face significant delays in recovery, leading to higher downtime and operational costs. Root cause analysis (RCA) addresses this gap by uncovering the underlying drivers of observed anomalies, disentangling direct causes from secondary effects, and enabling more targeted remediation Liu et al. (2023); Han et al. (2025). In complex cloud Nedelkoski et al. (2020) and cyber-physical environments Mathur & Tippenhauer (2016), where human monitoring alone is infeasible, automated RCA is essential for ensuring resilience and sustainable system management.

Root cause analysis (RCA) can be formally described as identifying, given a set of anomalous metrics, the top-$K$ metrics most likely responsible for the anomaly Liu et al. (2023). Unlike anomaly detection, which merely signals abnormal behavior, RCA requires interpretability: models must reveal how components influence one another and propagate faults across the system. Achieving this using statisical methods Ikram et al. (2022); Shan et al. (2019)is particularly challenging in modern infrastructures, where a single incident may involve thousands of KPIs, rendering manual tracing or heuristic correlations ineffective. Recent research has therefore shifted toward data-driven methods.

Among them, neural Granger causality Granger (1969) has emerged as a principled tool for uncovering temporal dependencies between variables, offering a systematic way to infer directional relationships. However, contemporary neural Granger causality methods Marcinkevičs & Vogt (2021b); **?** typically rely on MLPs applied independently at each time step. Such architectures prevent the model from capturing spatial dependencies across metrics, limiting its explainability across system components. Moreover, the per-time-step design also constrains the temporal horizon the model can consider and causes a parameter explosion as system dimensionality grows. Additionally, these approaches fail to account for seasonal and periodic patterns, which are crucial for understanding recurring system behaviors. These limitations highlight the need for more advanced RCA frameworks that can jointly model spatial and temporal dependencies while remaining interpretable and scalable to high-dimensional, real-world datasets.

A promising direction for RCA is to represent time series from multiple perspectives. Frequency-domain transformations have been shown to reveal latent structures that remain obscured in the raw time domain Xu et al. (2024); Yi et al. (2025; 2023). Hybrid approaches that jointly leverage temporal and frequency representations have demonstrated strong performance in anomaly detection Dou et al. (2025); Bai et al. (2023a). Despite these advances, integrating interpretability, a critical requirement for RCA, into multi-domain representations remains largely unexplored. We posit that combining time and frequency perspectives while explicitly enforcing interpretability can significantly enhance RCA. By moving beyond single-domain limitations, such approaches are better equipped to uncover the underlying mechanisms of complex anomalies in high-dimensional, large-scale monitoring systems.

In this work, we propose CrGSTA (Cross-domain Root causal Graph Spatial-Temporal Attention Network), a scalable and interpretable framework for root cause analysis in multivariate time series. CrGSTA is grounded in Granger causality Marcinkevičs & Vogt (2021b); Han et al. (2025); Fu et al. (2024), enabling unsupervised modeling of normal system behavior and the identification of exogenous factors that drive anomalies. Inspired by prior work on neural Granger causality Han et al. (2025) and cross-domain time- and frequency representations Dou et al. (2025); Bai et al. (2023a), CrGSTA captures complementary patterns across domains while enhancing the interpretability of detected anomalies. CrGSTA employs a spatio-temporal encoder–decoder architecture. The encoder features parallel time- and frequency-domain paths, each applying spatial graph attention across time lags followed by temporal attention. Their outputs are integrated via cross-attention, producing interpretable latent representations that reveal exogenous influences. A lightweight self-attention decoder reconstructs the series, and deviations from the learned normal distribution during inference are flagged as potential root causes, distinguished from downstream effects. Overall, CrGSTA offers a principled and scalable framework for multi-domain RCA in complex, high-dimensional systems by unifying cross-domain representation learning, spatio-temporal attention, and Granger causal reasoning.

Our experiments demonstrate that CrGSTA establishes a new state of the art for root cause analysis in multivariate time series by jointly modeling temporal and frequency domains through a graph-based encoder–decoder. Across both synthetic and real-world datasets, CrGSTA consistently outperforms statistical, non-causal, and causal deep learning baselines, while preserving parameter efficiency. For instance, it achieves 0.782 Avg@10 on Lotka–Volterra and 0.426 on SWaT, surpassing prior methods by wide margins despite operating under a fixed budget of only 8M parameters—more than two orders of magnitude fewer than AERCA's 200M+. Ablation studies further highlight the indispensability of cross-domain integration and attention mechanisms, which together enable CrGSTA to capture complex spatio-temporal dependencies without the prohibitive computational overhead observed in existing causal models. These findings not only validate the effectiveness of CrGSTA's architectural design but also underscore its practicality for large-scale monitoring systems where efficiency and interpretability are critical. In doing so, CrGSTA advances root cause analysis beyond current trade-offs between accuracy and scalability, pointing toward a new generation of resource-efficient causal modeling frameworks for modern infrastructures.

This work is guided by the following research questions: **RQ1:** How does CrGSTA perform as the temporal window size increases, and how does it compare to statistical and deep learning baselines in terms of accuracy and parameter efficiency? **RQ2:** How does CrGSTA scale with the number of interacting variables, and how does its performance and parameter growth compare to other deep learning approaches? **RQ3:** What are the contributions of CrGSTA's architectural components and fusion strategies to its overall performance, and how do they impact parameter efficiency? **RQ4:**

How effectively does CrGSTA capture complex causal relationships in real-world datasets, and what insights can be drawn from its interpretability features regarding root cause identification?

Our contributions are threefold: (1) We introduce CrGSTA, a novel unsupervised framework for root cause detection in multivariate time series that achieves a balance between scalability and interpretability, making it suitable for large-scale, complex real-world datasets. (2) We design a multi-path encoder–decoder architecture grounded in Granger causal reasoning, featuring parallel time- and frequency-domain paths. Spatial graph attention captures inter-variable dependencies, temporal self-attention models historical dynamics, and cross-attention fuses time- and frequency-domain representations, enabling the model to capture seasonality and periodic patterns. A lightweight self-attention decoder replaces conventional autoregressive stacks, resulting in substantial efficiency gains. (3) We perform extensive empirical evaluations on both synthetic and real-world datasets, systematically analyzing the impact of temporal and spatial dimensions as well as architectural choices, demonstrating the effectiveness and flexibility of CrGSTA in capturing complex causal relationships.

## 2 RELATED WORK

Root cause analysis (RCA) in multivariate systems intersects with performance engineering, where the goal extends beyond anomaly detection to scalable, interpretable, and robust diagnostics.

### 2.1 ROOT CAUSE ANALYSIS

RCA methods are broadly categorized into topology-driven, statistical, and causal inference–based approaches (Table 1). Topology-driven methods infer dependencies among variables and localize anomalies via graph traversal. For instance, MonitorRank Kim et al. (2013) scores service-level correlations using personalized PageRank Brin & Page (1998). While effective in structured environments, these methods often scale poorly in dynamic systems. Statistical techniques detect significant deviations in system metrics. $\epsilon$-Diagnosis Shan et al. (2019) employs two-sample tests, whereas RCD Ikram et al. (2022) applies conditional independence tests to infer causal structures. Although efficient and interpretable, these methods struggle with complex anomalies. Data-driven approaches learn temporal and spatial dependencies from multivariate observations Han et al. (2025); Tuli et al. (2022), and causal inference–based methods treat anomalies as interventions in structural causal models Assaad et al. (2022). CORAL Wang et al. (2023) incrementally updates a disentangled causal graph to capture both state-invariant and state-dependent dependencies, identifying root causes via network propagation in near-real time. GVAR Marcinkevičs & Vogt (2021b) uses self-explaining neural networks to infer Granger-causal relationships in multivariate time series, capturing nonlinear interactions and their temporal variability with interpretable causal effects. Building on GVAR, AERCA Han et al. (2025) leverages autoencoders to capture Granger causal dependencies. However, many existing designs rely on shallow parameterizations (e.g., MLP-based causal coefficients), limiting robustness in complex systems.

### 2.2 ORTHOGONAL ADVANCES IN TEMPORAL MODELING

Recent progress emphasizes lightweight yet expressive architectures, ranging from linear attention blocks to compact Transformers Tan et al. (2024); Liu et al. (2024). Frequency-domain methods have also proven highly efficient; for example, a 10K-parameter Fourier model matched the performance of a 300M-parameter Transformer Zhou et al. (2022); Xu et al. (2024), inspiring models such as FilterNet Yi et al. (2025), FourierGNN Yi et al. (2023), and FreqTimeLoss Wang et al. (2025a). Cross-domain architectures further enhance robustness by jointly leveraging temporal and spectral representations. CrossFuN Bai et al. (2023a) fuses temporal and spectral views, while DeAnomaly Dou et al. (2025) combines graph attention with time–frequency cross-attention to handle noisy multivariate data. These multi-domain approaches provide richer inductive biases than single-domain methods. Despite these advances, most anomaly detection models lack interpretability, and existing RCA approaches often rely on MLP-based Granger causality approximations that scale poorly and neglect temporal expressiveness. To address this gap, we propose CrGSTA, a spatio-temporal encoder–decoder that integrates time- and frequency-domain representations with graph-based causal reasoning, capturing long-range temporal dependencies and spatial interactions for scalable, interpretable RCA in complex multivariate systems.

## 3 PRELIMINARIES AND PROBLEM FORMULATION

Root cause analysis (RCA) in multivariate time series aims to identify latent factors driving observed variables. Granger causality Granger (1969) formalizes this: for a $d$-dimensional series $\{x_t\}_{t=1}^T$, each component $x_t^{(j)}$ can be expressed as a function of past values plus an unexplained latent input $z_t^{(j)}$,

$$x_t^{(j)} = f^{(j)}\big(x_{\leq t-1}^{(1)}, \ldots, x_{\leq t-1}^{(d)}\big) + z_t^{(j)}. \tag{1}$$

Here, $x^{(i)}$ Granger-causes $x^{(j)}$ if including its history improves prediction beyond $x^{(j)}$'s own past.

In an encoder–decoder view, the encoder extracts latent exogenous variables $z_t$ by removing predictable components, producing an interpretable representation of unexpected influences. The decoder reconstructs observations from these latent variables, ensuring consistency with the generative process. Formally, with $\mathbf{z}_t \in \mathbb{R}^d$ and $\mathbf{x}_t \in \mathbb{R}^p$, the marginal likelihood is

$$\log P(\mathbf{x}_t) = \log \int P(\mathbf{x}_t \mid \mathbf{z}_t, A(t))\, P(\mathbf{z}_t)\, d\mathbf{z}_t, \tag{2}$$

where $A(t)$ encodes instantaneous causal structure. The intractable posterior $P(\mathbf{z}_t \mid \mathbf{x}_t)$ is approximated by a variational distribution $E_\phi(\mathbf{z}_t \mid \mathbf{x}_{\leq t-1})$, yielding a VAE-like framework Kingma & Welling (2014). Graph attention captures cross-variable dependencies, temporal attention models sequential dynamics, and optional frequency-domain transformations reveal hidden patterns that improve interpretability.

RCA then identifies indices $(j, t)$ where latent variables deviate due to anomalies, $\hat{z}_t^{(j)} = z_t^{(j)} + \epsilon_t^{(j)}$. Unlike standard anomaly detection, the focus is on the sources of abnormal behavior.

### 3.1 CRGSTA WITH TIME-FREQUENCY CROSS-ATTENTION

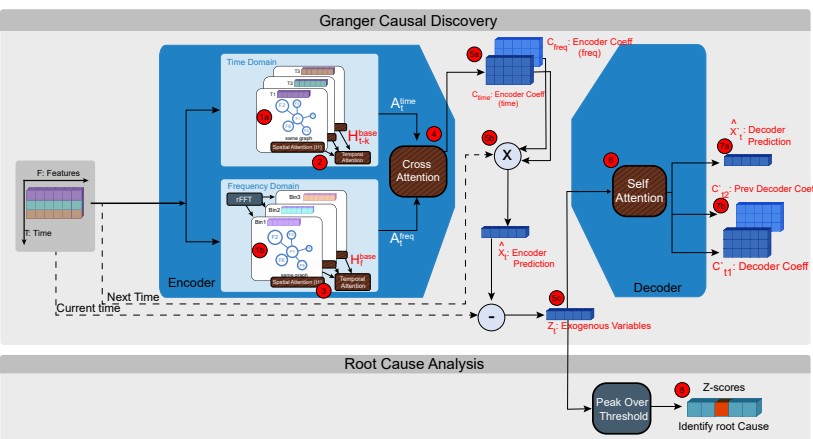

Figure 1: CrGSTA: Time-Frequency Cross-Attention Graph Spatio-Temporal Autoencoder

We present **CrGSTA** as a time-frequency cross-attention graph-based encoder-decoder for multivariate root cause identification, as illustrated in Fig. 1. Additionally, an intuitive summary of the architecture can be found in the Appendix A.3. The encoder estimates latent exogenous variables $E_\phi(\mathbf{z}_t \mid \mathbf{x}_{\leq t})$, while the decoder reconstructs the observation $\mathbf{x}_t$ given past exogenous sequences $D_\theta(\mathbf{x}_t \mid \mathbf{z}_{\leq t})$.

#### 3.1.1 ENCODER STRUCTURE

**Windowing Time Series.** Given $\mathbf{X} = (\mathbf{x}_1, \ldots, \mathbf{x}_T)$ with $d$ variables, we define sliding windows of length $K$:

$$\mathbf{W}_t = (\mathbf{x}_{t-K+1}, \ldots, \mathbf{x}_t), \quad \mathbf{W} = (\mathbf{W}_K, \ldots, \mathbf{W}_T), \tag{3}$$

so each window is processed to capture both temporal and spatial dependencies.

**Step 1: Base Spatial Graph (Shared Across Lags and Branches).** We define a global, shared graph attention network (GNN) to compute pairwise influence between variables. Each variable in a time step forms a node in a fully-connected graph. This shared graph serves as the foundation for both the time-domain and frequency-domain branches:

$$\mathbf{H}_{t-k}^{\text{base}} = \text{GNN}(\mathbf{x}_{t-k}) \in \mathbb{R}^{d \times d}, \quad k = 1, \dots, K \tag{4}$$

This design reduces parameter redundancy and ensures consistent modeling of interactions across domains.

**Step 2: Time-Domain Branch.** Using the shared base graph network, we apply temporal attention across lags to dynamically weight contributions of past observations:

$$\mathbf{A}_t^{\text{time}} = \text{TemporalAttn}([\mathbf{H}_{t-1}^{\text{base}}, \dots, \mathbf{H}_{t-K}^{\text{base}}]) \in \mathbb{R}^{K \times d \times d}. \tag{5}$$

**Step 3: Frequency-Domain Branch.** The shared base graph is also leveraged to capture frequency-domain dependencies. First, a real FFT is applied along the temporal axis to extract periodic components, yielding $\mathbf{X}_f^{\text{freq}} = \text{rFFT}(\mathbf{W}_t)_f$ for $f = 1, \dots, F$. The magnitudes of these frequency bins are then propagated through the shared graph network, followed by temporal attention across frequency bins:

$$\mathbf{H}_f^{\text{freq}} = \text{GNN}(|\mathbf{X}_f^{\text{freq}}|), \quad \mathbf{A}_t^{\text{freq}} = \text{TemporalAttn}([\mathbf{H}_1^{\text{freq}}, \dots, \mathbf{H}_F^{\text{freq}}]) \in \mathbb{R}^{F \times d \times d}. \tag{6}$$

**Step 4: Cross-Attention Fusion.** After obtaining temporal and spectral representations, we introduce explicit information exchange between the two modalities. Two cross-attention modules are employed: one aligns frequency features with temporal context (time→freq), while the other aligns temporal features with spectral context (freq→time). This bi-directional interaction yields the enriched representations $\tilde{\mathbf{H}}^{\text{time}}$ and $\tilde{\mathbf{H}}^{\text{freq}}$:

$$\tilde{\mathbf{H}}^{\text{time}} = \text{CrossAttn}(\mathbf{A}_t^{\text{time}}, \mathbf{A}_t^{\text{freq}}), \quad \tilde{\mathbf{H}}^{\text{freq}} = \text{CrossAttn}(\mathbf{A}_t^{\text{freq}}, \mathbf{A}_t^{\text{time}}). \tag{7}$$

**Step 5: Coefficient Projection and Prediction.** The cross-attended representations from Step 4 are projected through linear layers into adjacency-like coefficient matrices (step 5a), yielding

$$\mathbf{C}_{\text{time}} = \text{Linear}(\tilde{\mathbf{H}}^{\text{time}}), \quad \mathbf{C}_{\text{freq}} = \text{Linear}(\tilde{\mathbf{H}}^{\text{freq}}), \tag{8}$$

which encode variable-to-variable dependencies across lags $k$. Empirically, we find that constraining the *time-domain* coefficients is sufficient for stable optimization of the loss functions. Nevertheless, both the time and frequency coefficients contribute to autoregressive prediction (step 5b):

$$\hat{\mathbf{x}}_{\text{time}} = \sum_{k=1}^{K} \mathbf{C}_{\text{time}} \mathbf{x}_{t-k}, \quad \hat{\mathbf{x}}_{\text{freq}} = \sum_{k=1}^{K} \mathbf{C}_{\text{freq}} \mathbf{x}_{t-k}, \tag{9}$$

where $\mathbf{x}_{t-k}$ represents the historical observations within the input window. These modality-specific predictions are then combined linearly to produce the next-step prediction, which is also used to compute the residual relative to the current observation (step 5c):

$$\hat{\mathbf{x}}_t = \omega_t \hat{\mathbf{x}}_{\text{time}} + \omega_f \hat{\mathbf{x}}_{\text{freq}}, \quad \mathbf{z}_t = \mathbf{x}_t - \hat{\mathbf{x}}_t, \tag{10}$$

where $\omega_t$ and $\omega_f$ are the weights for combining both domains, and $\mathbf{z}_t$ is interpreted as a latent exogenous influence, capturing variability that is not explained by the temporal–spectral dynamics.

**Encoder Output.** In summary, the encoder produces two distinct outputs, each serving a specific purpose:

1. **Time-domain coefficients:** $\mathbf{C}_{\text{time}}$, which encode variable-to-variable dependencies and are directly used in the loss functions. These coefficients provide interpretability within the Granger-causal framework, as detailed in the subsequent sections.

2. **Latent exogenous variables:** $\mathbf{Z}_t \in \mathbb{R}^{d \times K}$, capturing influences not explained by the temporal–spectral dynamics, and serving as input to the decoder for reconstruction tasks.

### 3.1.2 DECODER STRUCTURE

The decoder reconstructs $\mathbf{x}_t$ from the exogenous sequence $\mathbf{Z}_t$ using a temporal-attention-based mechanism, avoiding fully autoregressive reconstruction.

**Step 6: Projection and Windowed Attention.** Each exogenous variable in the window is projected to a hidden representation $\mathbf{H}^{\text{enc}}_{t-K+\tau} = f_{\text{proj}}(\mathbf{z}_{t-K+\tau}), \quad \tau = 1, \ldots, K$, which are then aggregated via temporal attention across the window:

$$\mathbf{H}^{\text{temp}}_t = \text{TemporalAttn}(\mathbf{H}^{\text{enc}}_{t-K+1:t}), \tag{11}$$

producing a context-aware embedding for reconstruction.

**Step 7: Output and Low-Rank Coefficients.** The final prediction is obtained via a learnable output mapping $\hat{\mathbf{x}}_t = f_{\text{out}}(\mathbf{H}^{\text{temp}}_t)$, moreover generating low-rank coefficient matrices for interpretability:

$$\mathbf{C}_t = \mathbf{U}\mathbf{V}^\top, \quad \mathbf{C}_t \in \mathbb{R}^{d \times d}. \tag{12}$$

This structure efficiently captures temporal dependencies in the exogenous sequence while supporting interpretable causal attributions without maintaining separate decoders for past windows.

### 3.1.3 TRAINING OBJECTIVE

The encoder-decoder model is $\hat{\mathbf{x}}_t = \text{CrGSTA}_{\theta,\phi}(\mathbf{x}_{<t})$, with encoder parameters $\theta$ and decoder parameters $\phi$. For a series of length $T$, the training objective combines reconstruction, regularization and independence:

**Reconstruction Loss:** encourages the model to reconstruct the current step from latent exogenous variables:

$$\mathcal{L}_{\text{recon}} = \sum_{t=K+1}^{T} \|\hat{\mathbf{x}}_t - \mathbf{x}_t\|_2^2 \tag{13}$$

**Sparsity & Smoothness:** promote interpretable coefficient matrices in encoder and decoder:

$$\mathcal{L}_{\text{sparse}} = \lambda_{\text{enc}} R(\boldsymbol{\Omega}_t) + \lambda_{\text{dec}}\left(R(\bar{\boldsymbol{\Omega}}_t) + R(\bar{\boldsymbol{\Omega}}'_t)\right), \tag{14}$$

$$\mathcal{L}_{\text{smooth}} = \gamma_{\text{enc}} S(\boldsymbol{\Omega}_{t+1}, \boldsymbol{\Omega}_t) + \gamma_{\text{dec}}\left(S(\bar{\boldsymbol{\Omega}}_{t+1}, \bar{\boldsymbol{\Omega}}_t) + S(\bar{\boldsymbol{\Omega}}'_{t+1}, \bar{\boldsymbol{\Omega}}'_t)\right) \tag{15}$$

where $R$ imposes sparsity and $S$ enforces temporal smoothness. $\boldsymbol{\Omega}_t \in \mathbb{R}^{P \times P}$ is the encoder's time-varying coefficient (adjacency) matrix, with $\bar{\boldsymbol{\Omega}}_t$ and $\bar{\boldsymbol{\Omega}}'_t$ as decoder counterparts; we use $R(A) = \|A\|_1$ and $S(A, B) = \|A - B\|_F^2$.

**Exogenous Independence (KL):** encourages the latent exogenous variables $Z_t$ to be decorrelated and standardized:

$$\mathcal{L}_{\text{KL}} = \beta\, D_{\text{KL}}(P(\mathbf{Z}_t) \,\|\, Q) = \frac{1}{2}\left(\text{tr}(\Sigma_t) + \mu_t^\top \mu_t - d - \log\det \Sigma_t\right) \tag{16}$$

where $Q$ is an isotropic Gaussian prior.

**Total Objective:** the sum of all components:

$$\mathcal{L}_{\text{total}} = \mathcal{L}_{\text{recon}} + \mathcal{L}_{\text{sparse}} + \mathcal{L}_{\text{smooth}} + \mathcal{L}_{\text{KL}} \tag{17}$$

This formulation preserves interpretability, enforces latent independence, and supports the single-decoder CrGSTA architecture in reconstructing the time series while highlighting causal attributions.

### 3.2 ROOT CAUSE LOCALIZATION

**Step 8: Obtaining Root Causes:** During deployment, for a new observation $\mathbf{x}_{t^*}$, its exogenous representation $\mathbf{z}_{t^*}$ is computed using the trained encoder. Standardized scores (z-scores) are calculated:

$$z_{t^*}^{(j)} = \frac{z_{t^*}^{(j)} - \mu^{(j)}}{\sigma^{(j)}}, \tag{18}$$

and variables exceeding an adaptive threshold (via SPOT) are flagged as potential root causes.

## 4 EXPERIMENTS

### 4.1 DATASETS

To evaluate the effectiveness of the CrGSTA framework, we conduct experiments on four datasets: two synthetic benchmarks (Non-Linear and Lotka–Volterra) and two widely used real-world multivariate time series datasets (MSDS and SWaT) (Table 3, more details in appendix A.7.1). We extend the both of the synthetic datasets by increasing nonlinearity and stochastic variability, making

anomaly detection more challenging; full details of the extended model are provided in the appendix A.7.1. This controlled complexity allows for rigorous testing of root cause analysis methods under known causal structures.

## 4.2 Experimental Setup

**Baselines and Comparison.** We benchmark CrGSTA against statistical, non-causal, and causal deep learning approaches for root cause analysis. Among statistical baselines, $\epsilon$-Diagnosis Shan et al. (2019) uses pairwise significance tests, and RCD Ikram et al. (2022) identifies influential sources via partial causal graphs. For non-causal deep models, FEDformer Zhou et al. (2022) and iTransformer Liu et al. (2024) use frequency-enhanced or dual-domain attention and are adapted here by ranking variables through reconstruction errors. For causal deep learning, GVAR **?** models variable interactions using a graph-based encoder, causalrca Xin et al. (2023) incorporates Granger-inspired constraints into MLPs, and AERCA Han et al. (2025) employs lag-specific stacked MLPs for autoregressive reconstruction.

**Evaluation Metrics:** We evaluate root cause identification using the *recall at top-$k$* metric (AC@$k$) and its average variant (Avg@$k$), following prior work Ikram et al. (2022); Li et al. (2022b). This measures the likelihood that true root causes appear among the top-$k$ ranked variables. Sequences with multiple interventions are treated as single root cause sequences, consistent with point-adjust evaluation Koh et al. (2025); Bai et al. (2023b). Formal definitions are provided in the Appendix A.7.2. We also report the number of trainable parameters to assess efficiency, particularly for encoder–decoder models.

**Implementation:** We train multiple CrGSTA variants for each dataset variation, differing only in spatial–temporal attention dimension and attention heads. The decoder is identical for all datasets, using a lightweight self-attention layer with 64 hidden dimensions and 2 heads. Models are optimized with Adam (lr = 0.0001). Each experiment is repeated with multiple random seeds, and averages with standard deviations (reported in the appendix) ensure robustness. Experiments are run on a Linux workstation with an Intel i9-10900K CPU (20 cores, 3.70GHz), 32 GB RAM, and an NVIDIA RTX 3070 GPU (8 GB), using Python 3.10.12, PyTorch 2.7.1+cu126, and PyTorch Geometric 2.6.1. More details are in the Appendix A.8.

## 4.3 RQ1: Performance in Temporal Dimension

We evaluate CrGSTA's temporal scalability by varying the input window size, fixing the number of interacting variables to 40 for Lotka–Volterra, 20 for Non-Linear, and using all 51 variables for SWaT and 10 for MSDS. Results are shown in Fig. 2, with detailed analysis in Appendix A.9.1. **Lotka–Volterra.** Statistical methods remain largely flat (Avg@10 ≈ 0.16–0.18), highlighting their inability to capture nonlinear dependencies. Non-causal deep models show mild temporal sensitivity but saturate quickly: FEDformer peaks at window 5 (0.175), iTransformer at window 1 (0.166). Simple causal models such as causalrca with 256-unit MLPs plateau at 0.745 Avg@10. GVAR, due to its encoder-only architecture, cannot leverage longer windows effectively. AERCA benefits significantly from longer windows, improving from 0.584 (window 1) to 0.803 (window 5), albeit with substantial parameter growth (0.3M → 3.1M). In contrast, CrGSTA achieves the best accuracy under a fixed parameter budget (0.782 at window 7), with gains attributable to cross-domain temporal modeling rather than model size. **SWaT.** Statistical baselines remain below 0.2, while non-causal deep models reach only 0.315–0.334 without temporal scalability. Causalrca shows modest gains, peaking at 0.178 Avg@10 (window 5). GVAR improves with longer windows but incurs significant parameter growth. AERCA again benefits from causal modeling but exceeds 100M parameters—rendering it impractical, effectively a grey-box model that is over 10× larger than CrGSTA and prone to OOM or prohibitive training time at extreme windows. By contrast, CrGSTA achieves 0.426 Avg@10 at window 7—the best overall—while maintaining efficiency via cross-attention over medium-range dependencies. CrGSTA keeps a stable parameter count across window sizes (8.5M at window 7) compared to AERCA's 200M+, a two-orders-of-magnitude reduction. Additional analysis on Non-Linear and MSDS datasets (Appendix A.9.1) confirms these trends, with CrGSTA consistently achieving top performance with stable parameterization.

**Summary.** Statistical models fail to exploit temporal information; non-causal deep models capture limited temporal effects but saturate; causal models such as AERCA improve accuracy but incur prohibitive parameter costs. CrGSTA breaks this trade-off, achieving causal-level performance with stable parameterization, thereby highlighting the role of temporal–frequency interaction modeling in scalable root cause analysis.

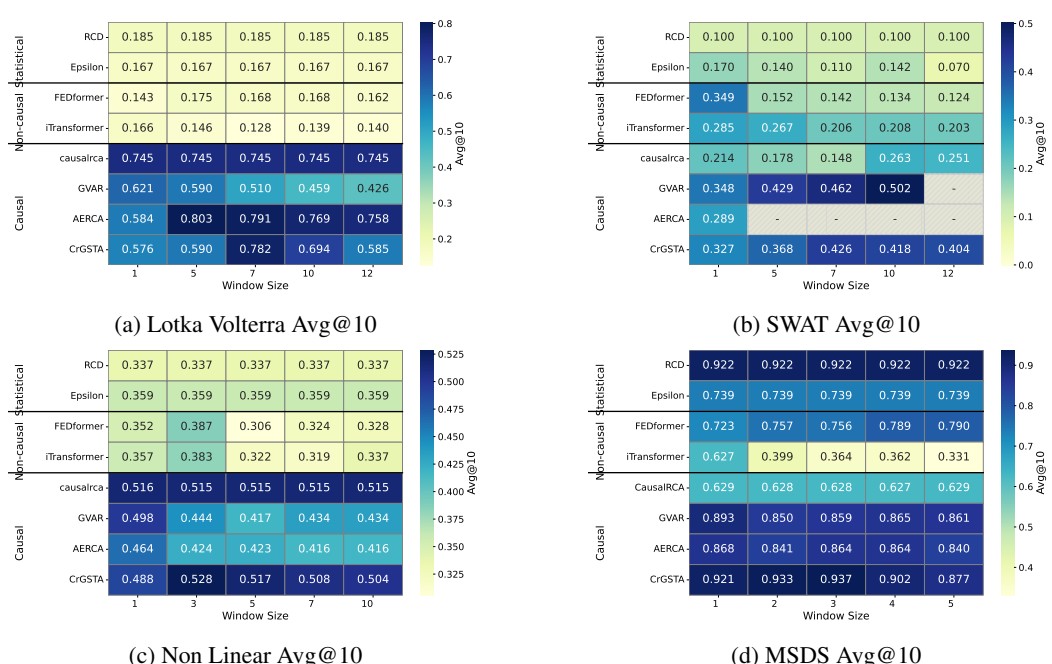

Figure 2: Performance (Avg@10) for different datasets.

### 4.4 RQ2: PERFORMANCE IN THE SPATIAL DIMENSION

To evaluate CrGSTA's spatial scalability, we fix the temporal window (7 for Lotka–Volterra, 5 for NonLinear) and vary the number of variables (Fig. 3; full tables in Appendix A.9.2). **Causal vs. Non-Causal Models.** Causal models clearly outperform non-causal baselines across all dimensionalities. At 20 variables, CrGSTA achieves the highest Avg@10 (0.866), far exceeding iTransformer (0.354) and FEDformer (0.272). At 60 variables, CrGSTA still maintains 0.755, while non-causal models collapse (iTransformer 0.126, FEDformer 0.103), underscoring the necessity of causal modeling in high-dimensional systems. Although causalrca performs competitively at small scales, it is consistently outperformed by AERCA and CrGSTA and degrades more rapidly as dimensionality increases. **Parameter Efficiency.** CrGSTA matches or exceeds the performance of larger causal models while using substantially fewer parameters. For example, at 20 variables it achieves 0.866 Avg@10 with 0.4M parameters (AERCA: 0.859 with 0.5M), and at 50 variables it reaches 0.734 with 1.3M (AERCA: 0.749 with 2.8M). Even at 60 variables, CrGSTA sustains 0.755 using only 1.7M parameters. Its parameter growth is limited to lightweight per-variable adapters, keeping the attention dimension fixed and ensuring linear, scalable complexity. **Consistency Across Datasets.** The same patterns appear on the NonLinear dataset (more details in Appendix A.9.2).

**Summary.** CrGSTA achieves robust high-dimensional causal performance with notable parameter efficiency. On Lotka–Volterra, it maintains over 75% Avg@10 at 60 variables while using less than half the parameters of AERCA, demonstrating strong scalability for complex dynamical systems.

### 4.5 RQ3: ABLATION STUDIES

We evaluate CrGSTA's components by varying spatial architectures and fusion strategies, fixing the temporal window to 7 and using 40 variables for Lotka–Volterra. To highlight architectural differences, we set the attention dimension to 32 on Lotka–Volterra and 256 on SWaT. Results and details

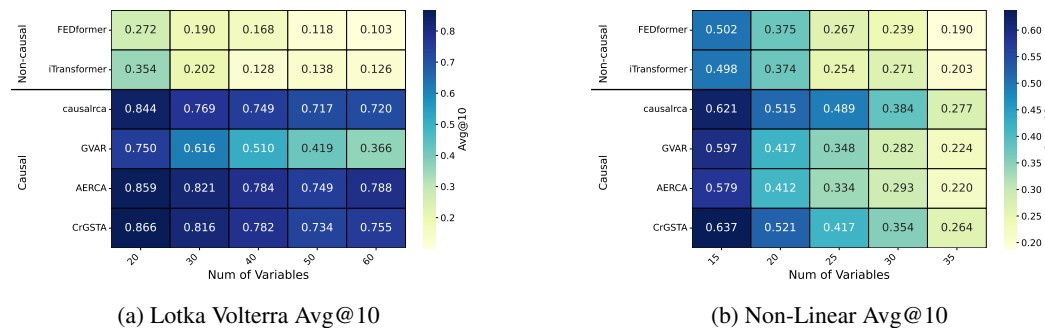

(a) Lotka Volterra Avg@10

(b) Non-Linear Avg@10

Figure 3: Performance (Avg@10) for Lotka Volterra (left) and Non-Linear (right).

on the ablation configurations are shown in Fig. 4 and Tables 14, 15 in the Appendix. **Spatial Architectures.** On Lotka–Volterra, temporal-only models (T) perform well (Avg@10=0.546). On SWaT, however, frequency-only models (F) surpass temporal-only ones (0.365 vs. 0.334), highlighting the importance of frequency features in complex systems. **Frequency Representations.** Interstingly, For SWaT, the frequency-only (F (Mag)) model outperforms temporal-only (T) (0.365 vs. 0.334), indicating that frequency features may better capture anomalies in complex data. This suggests that even before fusion, frequency-domain analysis can be more informative than time-domain for certain real-world systems. **Fusion Strategies.** For synthetic data, combining temporal and frequency features (T-F) with simple fusion (sum, concat, gated) gives moderate gains (Avg@10=0.525–0.575). On SWaT, these methods underperform frequency-only models (0.334–0.360), suggesting naive fusion adds redundancy. By contrast, CrGSTA's cross-domain attention (T-F with attn) achieves the best results on both datasets (0.639 for Lotka–Volterra, 0.430 for SWaT), showing the effectiveness of adaptive integration. **Magnitude vs. Magnitude–Phase.** Magnitude-only features often match or outperform magnitude–phase. On Lotka–Volterra, both achieve Avg@10=0.639. On SWaT, magnitude-only slightly outperforms (0.430 vs. 0.425), suggesting phase may add noise slight in complex data. **Parameter Efficiency.** CrGSTA with attention fusion is compact (1.0M params on Lotka–Volterra, 8.5M on SWaT) compared to concat (5.5M and 21.5M+), confirming that gains stem from cross-domain design rather than size.

**Summary.** CrGSTA's strengths come from attention and cross-domain integration, enabling accurate and efficient root cause analysis.

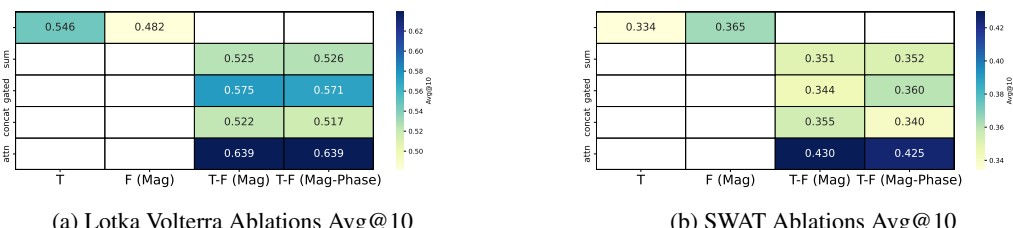

(a) Lotka Volterra Ablations Avg@10

(b) SWAT Ablations Avg@10

Figure 4: Architectural and Combinatorial Ablations for Lotka Volterra (a) and SWAT (b).

### 4.6 RQ4: Case Studies

To illustrate CrGSTA's interpretability, we present case studies on MSDS. Specifcally, we visualize two root cause analysis examples comparing CrGSTA and AERCA. First, we show the root cause scores assigned by both models for a specific anomaly instance. Second, we analyze the temporal propagation of root cause scores across multiple time steps within an analysis window.

#### 4.6.1 Root Cause Z-score

As shown in Fig. 5, the true root cause variable is highlighted with a red box. As illustrated in Fig. 5b CrGSTA accurately identifies the true root cause variable with a significantly higher score than

other variables, demonstrating its effectiveness in root cause analysis. In contrast, AERCA assigns relatively lower scores to the true root cause variable, indicating less confidence in its identification. This comparison highlights CrGSTA's superior interpretability and precision in pinpointing root causes within complex multivariate time series data. More details in the appendix A.9.4.

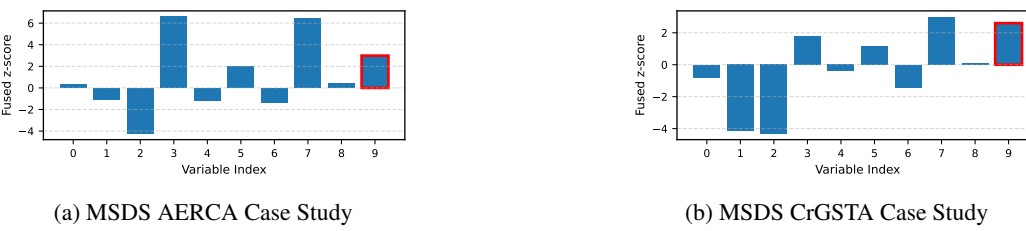

(a) MSDS AERCA Case Study                    (b) MSDS CrGSTA Case Study

Figure 5: Case Studies for MSDS Dataset: (a) AERCA (b) CrGSTA.

### 4.6.2 ROOT CAUSE Z-SCORE PROPAGATION THROUGH TIME

To further demonstrate the superior robustness and parameter efficiency of CrGSTA, we conduct a detailed temporal analysis case study. As depicted in Figure 6, we utilize an analysis window size ($W$) of 3 for both AERCA and CrGSTA on the MSDS dataset, focusing on a specific anomaly instance. The results illustrate CrGSTA's superior temporal consistency: it maintains a persistently high Normalized z-score across the window steps for the true root cause variable (index 9), strongly following the ground-truth label. In contrast, the baseline AERCA exhibits a scattered score distribution, failing to sustain a high score specifically for the root cause, leading to high ambiguity. This performance difference is particularly notable considering the significant disparity in model complexity: AERCA has over 28 million parameters, while CrGSTA operates with only 69,129 parameters. This underscores CrGSTA's ability to achieve more reliable, high-confidence root cause identification with dramatically fewer resources across varying temporal contexts.

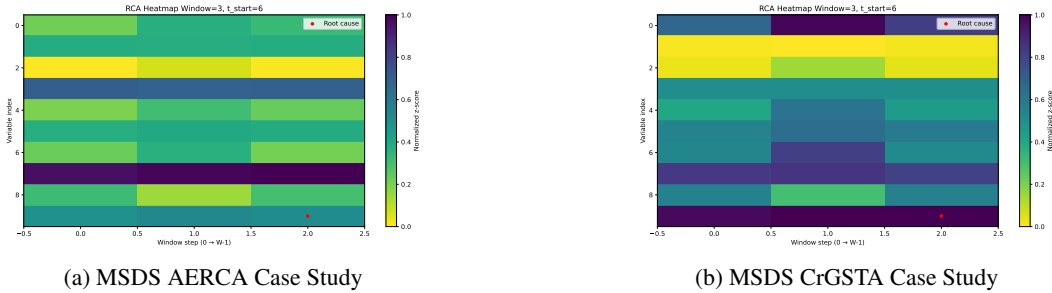

(a) MSDS AERCA Case Study                    (b) MSDS CrGSTA Case Study

Figure 6: Case Studies for MSDS Dataset: (a) AERCA (b) CrGSTA.

## 5 CONCLUSION

We introduced CrGSTA, a novel framework for root cause analysis in multivariate time series that effectively integrates temporal and frequency domain information through a graph-based encoder-decoder architecture with cross-attention. Extensive experiments on both synthetic and real-world datasets demonstrate that CrGSTA consistently outperforms statistical methods, non-causal models, and other causal deep learning baselines in terms of accuracy and scalability. Ablation studies further highlight the critical role of attention mechanisms, cross-domain integration, and architectural design in enabling precise root cause identification. Importantly, CrGSTA achieves these gains while maintaining parameter efficiency, making it well-suited for practical deployment, whereas other causal models often entail prohibitive computational costs. For future work, we plan to explore extending CrGSTA with *state space models* such as Mamba to complement or replace attention mechanisms, which could enhance long-horizon temporal reasoning and mitigate quadratic scaling. We also aim to investigate integrating multimodal data sources, such as metrics and logs, and how to overcome the challenges of combining these heterogeneous signals for effective root cause analysis.

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

## A APPENDIX

### A.1 RELATED WORKS

In this section we provide a comparison of various Root Cause Analysis (RCA) and Anomaly Detection approaches in Table 1. The table categorizes methods based on their graph structure, attention mechanisms, interpretability, and key strengths. This comparison highlights the unique features and advantages of each approach, providing a comprehensive overview of the landscape in this research area.

Table 1: Related Works Comparison of Generic Time Series Models and Root Cause Analysis Methods

| Method | Graph Structure | Attention | Interpretable | Key Strengths |
|---|---|---|---|---|
| **Generic Time Series Models** | | | | |
| **Time Domain** | | | | |
| iTransformer Liu et al. (2024) | ✗ | ✓(Linear Self-Attn) | ✗ | Efficient for long sequences; scalable forecasting |
| **Frequency Domain** | | | | |
| FEDformer Zhou et al. (2022) | ✗ | ✓(Sparse Fourier Attn) | ✗ | Captures periodic patterns; reduced complexity |
| FITS Xu et al. (2024) | ✗ | ✗(Frequency MLP) | ✗ | High-resolution freq modeling; compact design |
| **Time–Frequency Domain** | | | | |
| CrossFuN Bai et al. (2023a) | ✗ | ✗(simple Time–Freq fusion) | ✗ | Fuses temporal and spectral info |
| DeAnomaly Dou et al. (2025) | ✓(Graph) | ✓(Cross Time–Freq Attn) | ✗ | Robust to noise; joint graph + time–freq fusion |
| **Root Cause Analysis Models** | | | | |
| **Topology-Based Graph Methods** | | | | |
| MonitorRank Kim et al. (2013) | ✓(Call Graph) | ✗ | ✗ | PageRank-style ranking; interpretable |
| MicroRCA Wu et al. (2020) | ✓(Topology) | ✗ | ✗ | Random walk scoring on anomalous subgraphs |
| **Classical Statistical Techniques** | | | | |
| $\epsilon$-Diagnosis Shan et al. (2019) | ✗ | ✗ | ✗ | Lightweight; interpretable; efficient |
| N-Sigma Li et al. (2022a) | ✗ | ✗ | ✗ | Simple thresholding; effective for small anomalies |
| BARO Landsittel et al. (2020) | ✗ | ✗ | ✗ | Bayesian change-point detection; robust scoring |
| **Causal Inference and Graph Neural Methods** | | | | |
| GVAR Marcinkevičs & Vogt (2021b) | ✗ | ✗(Time MLP) | ✓ | Infers nonlinear Granger causality; interpretable causal effects in time series |
| CORAL Wang et al. (2023) | ✓(Disentangled Causal Graph) | ✗ | ✓ | Incremental online RCA; captures state-invariant and dependent dependencies |
| AERCA Han et al. (2025) | ✗ | ✗(Time MLP) | ✓ | Models interventions; interpretable |
| Ours (CrGSTA) | ✓(Graph Attn) | ✓(Spatio-Temporal Cross Time-Freq Attn) | ✓ | Scalable; captures long-range dependencies; hybrid domain; GNN+Attn |

## A.2 MODEL

In this section, we provide a detailed illustration of our proposed CrGSTA architecture in Figure 7. This figure visually represents the key components and data flow within the model, highlighting the loss functions, and the same graph structure used across time slices for clarity, and similarly another single graph used across frequency slices.

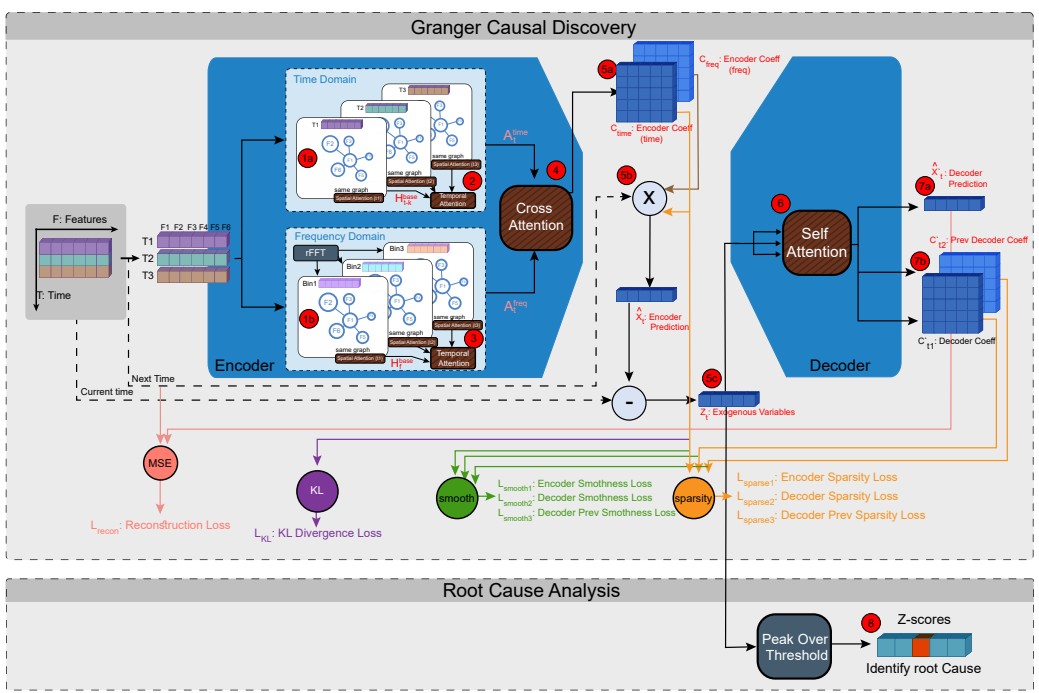

Figure 7: CrGSTA: Time-Frequency Cross-Attention Graph Spatio-Temporal Autoencoder

## A.3 Intuitive Explanation of Model Equations

### A.3.1 Encoder Equations

**Windowing.**

$$\mathbf{W}_t = (\mathbf{x}_{t-K+1}, \ldots, \mathbf{x}_t) \tag{19}$$

This extracts a sliding window of past observations. This allows the model to capture temporal dependencies over the recent history of the multivariate time series.

**Base Spatial Graph.**

$$\mathbf{H}_{t-k}^{\text{base}} = \text{GNN}(\mathbf{x}_{t-k}) \tag{20}$$

This computes pairwise interactions between variables at each lag. Using a graph neural network (GNN) enables the model to learn complex relationships among variables, which is crucial for identifying causal influences in multivariate data.

**Time-Domain Attention.**

$$\mathbf{A}_t^{\text{time}} = \text{TemporalAttn}([\mathbf{H}_{t-1:\,t-K}^{\text{base}}]) \tag{21}$$

This assigns attention weights to past time steps. This temporal attention mechanism allows the model to focus on the most relevant historical information when inferring exogenous variables.

**Frequency-Domain Attention.**

$$\mathbf{A}_t^{\text{freq}} = \text{TemporalAttn}([\mathbf{H}_1^{\text{freq}}, \ldots, \mathbf{H}_F^{\text{freq}}]) \tag{22}$$

This assigns attention weights to frequency components. Studying the frequency domain helps capture periodic patterns and anomalies that may not be evident in the time domain alone. Similar to temporal attention, this mechanism helps the model identify important spectral features that may indicate anomalies or causal factors.

**Cross-Attention Fusion.**

$$\tilde{\mathbf{H}}^{\text{time}} = \text{CrossAttn}(\mathbf{A}_t^{\text{time}}, \mathbf{A}_t^{\text{freq}}) \tag{23}$$

This mixes information between time and frequency. Mixing both the temporal and spectral representations allows the model to leverage complementary information from both domains, enhancing its ability to detect anomalies and infer causal relationships.

**Coefficient Projection**

**Projection.**

$$\mathbf{C}_{\text{time}} = \text{Linear}(\tilde{\mathbf{H}}^{\text{time}}) \tag{24}$$

This converts fused features into coefficient matrices. Specifcally, the coefficients represent variable-to-variable dependencies across lags. This helps identify causal influences between variables (i.e., sensors or metrics such as how cpu usage affects memory usage over time).

**Lagged Prediction.**

$$\hat{\mathbf{x}}_{\text{time}} = \sum_{k=1}^{K} \mathbf{C}_{\text{time}} \, \mathbf{x}_{t-k} \tag{25}$$

This predicts the next value using past windows. This approach captures autoregressive relationships, allowing the model to forecast future observations based on learned dependencies.

**Combination and Residual.**

$$\mathbf{z}_t = \mathbf{x}_t - \hat{\mathbf{x}}_t \tag{26}$$

This computes the deviation from prediction. This computed z helps isolate exogenous factors not explained by past values. Subsecuntly, these exogenous variables are used by the decoder to reconstruct the original observations.

A.3.2    DECODER EQUATIONS

**Temporal Attention on Exogenous Inputs.**

$$\mathbf{H}_t^{\text{temp}} = \text{TemporalAttn}(\mathbf{H}_{t-K+1:t}^{\text{enc}}) \tag{27}$$

This aggregates information across the window. By attending to the sequence of inferred exogenous variables, the model captures temporal dependencies that are essential for accurate reconstruction of the original observations.

**Reconstruction.**

$$\hat{\mathbf{x}}_t = f_{\text{out}}(\mathbf{H}_t^{\text{temp}}) \tag{28}$$

This produces the reconstructed output. The function $f_{\text{out}}$ maps the aggregated exogenous information back to the observation space, enabling the model to reconstruct the original multivariate time series.

**Low-Rank Coefficients.**

$$\mathbf{C}_t = \mathbf{U}\mathbf{V}^{\top} \tag{29}$$

This builds a low-rank interaction matrix. Specifcally, the low-rank structure encourages simpler, more interpretable relationships between exogenous variables and observations.

A.4    TRAINING OBJECTIVES

**Reconstruction Loss.**

$$\mathcal{L}_{\text{recon}} = \sum_t \|\hat{\mathbf{x}}_t - \mathbf{x}_t\|_2^2 \tag{30}$$

This penalizes reconstruction errors. For instance, minimizing this loss ensures that the model accurately captures the underlying data distribution, which is crucial for effective root cause analysis.

**Sparsity.**

$$\mathcal{L}_{\text{sparse}} = \lambda_{\text{enc}} R(\boldsymbol{\Omega}_t) + \lambda_{\text{dec}} (R(\bar{\boldsymbol{\Omega}}_t) + R(\bar{\boldsymbol{\Omega}}'_t)) \tag{31}$$

This encourages sparse coefficients. Specifcally, this promotes simpler causal structures by penalizing unnecessary connections between variables, making it easier to identify key root causes.

**Smoothness.**

$$\mathcal{L}_{\text{smooth}} = \gamma_{\text{enc}} S(\boldsymbol{\Omega}_{t+1}, \boldsymbol{\Omega}_t) + \gamma_{\text{dec}} (S(\bar{\boldsymbol{\Omega}}_{t+1}, \bar{\boldsymbol{\Omega}}_t) + S(\bar{\boldsymbol{\Omega}}'_{t+1}, \bar{\boldsymbol{\Omega}}'_t)) \tag{32}$$

This encourages coefficient stability across time. This loss ensures that the inferred relationships do not fluctuate wildly between consecutive time steps, which is important for maintaining consistent causal interpretations over time.

**KL Term.**

$$\mathcal{L}_{\text{KL}} = \frac{1}{2} \left( \text{tr}(\Sigma_t) + \mu_t^\top \mu_t - d - \log \det \Sigma_t \right) \tag{33}$$

This regularizes the latent space. By constraining the distribution of latent exogenous variables, this loss helps prevent overfitting and encourages the model to learn meaningful representations that generalize well to unseen data.

**Total Loss.**

$$\mathcal{L}_{\text{total}} = \mathcal{L}_{\text{recon}} + \mathcal{L}_{\text{sparse}} + \mathcal{L}_{\text{smooth}} + \mathcal{L}_{\text{KL}} \tag{34}$$

This is the final training objective.

### A.5 ROOT CAUSE LOCALIZATION

$$z_{t^*}^{(j)} = \frac{z_{t^*}^{(j)} - \mu^{(j)}}{\sigma^{(j)}} \tag{35}$$

This standardizes the exogenous variables for anomaly scoring. This approach highlights variables that deviate significantly from their normal behavior, aiding in the identification of potential root causes.

### A.6 JUSTIFICATION OF ARCHITECTURAL CHOICES.

Each module in Fig. 1 is introduced to align the model's computations with the underlying Granger-causal design.

**Motivation** Rootcause analysis in multivariate time series requires capturing complex interactions among variables over time. The architectural choices in CrGSTA are motivated by the need to effectively model these interactions while ensuring interpretability and scalability. While prior works have explored interpetability using Granger Causality Marcinkevičs & Vogt (2021b); Han et al. (2025), their architectural choice lack the scalability of modeling long-range dependencies and the ability to capture both time and frequency domain information. Specifcally, both these prior works rely on separate MLP per lag, while this allows for the interpretability of the learned coefficients, it easily becomes intractable when the number of lags increases.

**Graph Stduture.** The graph structure captures variable interactions, essential for modeling causal relationships. This was motivated by prior work demonstrating the effectiveness of graph-based methods for multivariate time series analysis Wang et al. (2025b); Huang et al. (2023). This desgin choice allows for a better representation of the dependencies among variables, which is crucial for accurate root cause identification.

**Frequency Domain.** Incorporating frequency-domain information enables the model to capture periodic patterns and anomalies that may not be evident in the time domain alone. This dual-domain approach enhances the model's ability to detect anomalies and infer causal relationships, as certain anomalies may manifest more clearly in the spectral representation. This was inspired by recent success of prior works on frequency-based time series modeling Xu et al. (2024); Wang et al. (2025a).

**Spatial and Temporal Attention.** The spatial attention mechanism allows the model to focus on the most relevant variables at each time step, while the temporal attention captures dependencies across time. This desgin choice allows the model to effectively learn complex interactions in both space and time, which is essential for accurate root cause analysis. Additionally, it allows for a more parameter efficient representation compared to using separate attention mechanisms for each variable or time step while allowing for an interpretable attention weights that can be used for root cause identification. This has been shown to be effective in prior works Huang et al. (2023).

**Encoder-Decoder Structure.** The encoder-decoder architecture separates the tasks of inferring exogenous variables and reconstructing observations. This enables the model to learn a more structured representation of the data, facilitating the identification of root causes. It builds on the success

of prior works Han et al. (2025); Wang et al. (2025b) that have demonstrated the effectiveness of this structure for anomaly detection and root cause analysis.

**Cross-Attention Fusion.** This innovative module allows the model to integrate information from both time and frequency domains. It has been motivated by prior works Bai et al. (2023a); Dou et al. (2025) that have shown the benefits of combining temporal and spectral information for improved anomaly detection.

Table 2: CrGSTA Modules, Their Inspirations, and Motivations

| Module | Inspiring Works | Why It Was Used |
|---|---|---|
| Graph Structure (Shared GNN) | Wang et al. (2025b); Huang et al. (2023) | Captures variable interactions and causal dependencies; provides a scalable structural prior replacing per-lag MLPs; improves interpretability and stability across domains. |
| Frequency-Domain Module | Xu et al. (2024); Wang et al. (2025a) | Extracts periodic and oscillatory patterns not visible in raw time series; improves anomaly detection sensitivity and complements time-domain reasoning. |
| Spatial & Temporal Attention | Huang et al. (2023) | Learns variable relevance and long-range temporal structure; more parameter-efficient than per-variable attention; provides interpretable weights for RCA. |
| Encoder–Decoder Structure | Han et al. (2025); Wang et al. (2025b) | Separates exogenous-variable inference from reconstruction; produces more structured latent representations and improves causal interpretability during anomalies. |
| Cross-Attention Fusion | Bai et al. (2023a); Dou et al. (2025) | Enables bidirectional fusion between temporal and spectral representations; captures anomalies that manifest differently across domains; enhances robustness. |

## A.7 Evaluation Datasets and Metrics

### A.7.1 Dataset

Table 3: Statistics of datasets.

| Dataset | Training Steps | Test Sequences ($|X|$) | Avg. Length ($T$) | Avg. Root Vars |
|---|---|---|---|---|
| **Synthetic Datasets** | | | | |
| Nonlinear (20) Marcinkevičs & Vogt (2021a) | 5,000 | 100 | 500 | 5.25 |
| Lotka–Volterra (40) Marcinkevičs & Vogt (2021a) | 40,000 | 100 | 2,000 | 30.75 |
| **Real-World Datasets** | | | | |
| SWaT (51) Mathur & Tippenhauer (2016) | 49,500 | 20 | 51 | 13.35 |
| MSDS (10) Nedelkoski et al. (2020) | 29,268 | 4,255 | 21 | 3.05 |

**Lotka–Volterra (Extended).** Extending the work of Marcinkevičs & Vogt Marcinkevičs & Vogt (2021a) and its implementation in Han et al. (2025), we introduce additional nonlinearities, stochastic variability, and more realistic adversarial perturbations. Instead of the original formulation

$$\frac{dx^{(i)}}{dt} = \alpha x^{(i)} - \beta \sum_{j \in Pa(x^{(i)})} y^{(j)} - \eta \left(x^{(i)}\right)^2, \tag{36}$$

$$\frac{dy^{(j)}}{dt} = \delta y^{(j)} \sum_{k \in Pa(y^{(j)})} x^{(k)} - \rho y^{(j)}, \tag{37}$$

$$x_t^{(i)} = x_t^{(i)} + 10\,\epsilon_t^{(i)}, \quad y_t^{(j)} = y_t^{(j)} + 10\,\epsilon_t^{(j)}, \quad 1 \le i, j \le p, \tag{38}$$

we build the extended version as

$$\frac{dx^{(i)}}{dt} = \alpha x^{(i)} - \beta \sum_{j \in Pa(x^{(i)})} y^{(j)} - \eta\big(x^{(i)}\big)^2 + \cos\big(x^{(i)}+1\big) + 0.5\sin\big(x^{(i)}\big) + \sigma\mathcal{N}(0,1), \quad (39)$$

$$\frac{dy^{(j)}}{dt} = \delta y^{(j)} \sum_{k \in Pa(y^{(j)})} x^{(k)} - \rho y^{(j)} + \cos\big(y^{(j)}+1\big) + 0.5\sin\big(y^{(j)}\big) + \sigma\mathcal{N}(0,1), \quad (40)$$

$$x_t^{(i)} = x_t^{(i)} + 2\,\epsilon_t^{(i)}, \quad y_t^{(j)} = y_t^{(j)} + 2\,\epsilon_t^{(j)}, \quad 1 \le i,j \le p. \quad (41)$$

Here, $x^{(i)}$ and $y^{(j)}$ denote prey and predator populations, respectively; $\alpha, \beta, \eta, \delta, \rho$ are interaction parameters; $\sigma$ introduces stochastic fluctuations; and $\epsilon_t^{(\cdot)}$ represents adversarial perturbations. By replacing the anomaly multiplier of 10 with 2 and enriching the dynamics with sinusoidal and noise terms, the anomalies become more subtle and thus better reflect realistic system behavior. Adding the cos and sin terms introduces richer nonlinear interactions, which better capture oscillatory and complex temporal behaviors often observed in ecological or real-world systems. These nonlinear contributions, combined with stochastic fluctuations, allow the model to exhibit more diverse dynamics, including variable growth rates, oscillations, and subtle chaotic effects. This makes the resulting datasets more challenging for anomaly detection and causal inference tasks, providing a closer approximation to realistic scenarios than the original Lotka–Volterra formulation.

**Non–Linear (Extended).** Extending standard synthetic autoregressive formulations and its implementation in Han et al. (2025), we construct a significantly richer nonlinear generator that incorporates higher-order temporal dependencies, expressive nonlinear interactions, correlated stochastic noise, and a causal anomaly mechanism. Whereas the baseline model follows a simple linear recurrence,

$$x_t^{(i)} = \sum_{l=1}^{L} A_{i:}^{(l)}\, x_{t-l} + \epsilon_t^{(i)}, \quad (42)$$

our extended formulation introduces nonlinearity, lag-specific mixing, and structured variability:

$$x_t^{(i)} = \sum_{l=1}^{5} A_{i:}^{(l)}\Big(\sin\big(x_{t-l}+0.5\big) + \log\big(1+|x_{t-l}|\big)\Big) + \epsilon_t^{(i)}. \quad (43)$$

Here each $A^{(l)}$ is obtained by scaling the underlying causal adjacency matrix with lag-dependent coefficients, enabling distinct temporal effects across the previous five time steps. The combination of $\sin(\cdot)$ and $\log(1+|\cdot|)$ introduces oscillatory, amplitude-dependent, and mildly chaotic behavior, substantially increasing the complexity of the normal data compared to classical linear autoregressive processes.

We also introduce a causal anomaly mechanism that allows perturbations to persist and propagate through the system. For non-causal anomalies, deviations are added directly and instantaneously to the noise:

$$\epsilon_t^{(i)} \leftarrow \epsilon_t^{(i)} + a^{(i)}, \qquad t \in \mathcal{T}_{\text{anom}}. \quad (44)$$

To model anomalies that spread through the nonlinear temporal dynamics, we introduce a latent anomaly effect $z_t$ with exponential decay:

$$z_t = 0.95\,z_{t-1} + a, \qquad\qquad t \in \mathcal{T}_{\text{anom}}, \quad (45)$$
$$z_t = 0.95\,z_{t-1}, \qquad\qquad t \notin \mathcal{T}_{\text{anom}}, \quad (46)$$
$$\epsilon_t \leftarrow \epsilon_t + z_t. \qquad\qquad (47)$$

Injecting the perturbation into the noise—rather than directly modifying the state—forces the anomaly to propagate through the nonlinear transformation and multi-lag mixing structure, producing cascading effects that resemble fault propagation in realistic systems.

### A.7.2 EVALUATION METRICS

**Recall at Top-$k$ (AC@$k$).**  Following prior work Ikram et al. (2022); Li et al. (2022b), we evaluate root cause identification using the *recall at top-$k$* metric, denoted AC@$k$. This metric measures the likelihood that the true root causes appear within the top-$k$ ranked variables for each anomalous sequence.

Formally, let $X \in \mathcal{X}$ denote an anomalous sequence, $R_X[k]$ the top-$k$ ranked variables produced by the model, and $V_X^{(\text{RC})}$ the ground-truth root cause set. Then,

$$\text{AC@}k = \frac{1}{|\mathcal{X}|} \sum_{X \in \mathcal{X}} \frac{\left| V_X^{(\text{RC})} \cap \{R_X[1], \ldots, R_X[k]\} \right|}{\min(k, |V_X^{(\text{RC})}|)}. \tag{48}$$

This definition ensures normalization when multiple root causes exist, by dividing by $\min(k, |V_X^{(\text{RC})}|)$.

**Average Recall (Avg@$k$).**  To summarize overall performance across different cutoffs, we also report the averaged metric:

$$\text{Avg@}k = \frac{1}{k} \sum_{i=1}^{k} \text{AC@}i. \tag{49}$$

This provides a more comprehensive measure than a single cutoff.

**Multiple Interventions.**  When a sequence contains multiple exogenous interventions, we treat it as a single root cause sequence, following the *point-adjust evaluation* protocol Koh et al. (2025); Bai et al. (2023b). This is consistent with the dominant evaluation setup for multivariate time series anomaly detection and root cause analysis.

**Model Efficiency.**  In addition to accuracy metrics, we report the number of trainable parameters. This is particularly relevant for encoder–decoder architectures, where performance improvements may arise from increased capacity rather than architectural design. Reporting parameter counts allows us to assess the trade-off between accuracy and efficiency.

### A.8 IMPLEMENTATION DETAILS

In this section, we summarize the key configurations used in our experiments (Tables 5, 4, and 6 and 7); full details are available in our released code. For AERCA, we adopt the original implementation Han et al. (2025) with its reported hyperparameters. For our CrGSTA model, we set the spatial–temporal attention dimension to 64 on Lotka–Volterra and 256 on SWaT, with 2 attention heads in both cases, we then use the same dimensions for other attention based baselines for a fair comparison. This approach is followed for Nonlinear and MSDS (albeit with different attention dimensions due to dataset size). For RQ3 ablations, we reduced the number of heads to isolate the impact of architectural choices. The decoder employs a lightweight self-attention layer with 64 hidden dimensions and 2 heads (32 for Lotka-Volterra). All models are trained with Adam (learning rate $10^{-4}$).

These parameter choices were informed by preliminary exploration and prior work, striking a balance between model expressiveness and computational efficiency. Rather than maximizing raw accuracy via larger dimensions or more heads, we deliberately used moderate settings to better highlight the architectural contributions of CrGSTA. Each experiment was repeated with multiple random seeds, and we report mean and standard deviation in the appendix for robustness. We further mention the motivations behind certain hyperparameter choices below after the tables.

Table 4: Experiment Configurations for Lotka–Volterra Benchmark

| Key Parameter | FEDformer ¶ | iTransformer ¶ | CuasalRCA† | AERCA* GVAR** | CrGSTA |
|---|---|---|---|---|---|
| Learning Rate | 1e-4 | 1e-4 | 1e-4 | 1e-4 | 1e-4 |
| Attention Dim | 64 | 64 | – | – | (spatial 64) (temporal 64) (decoder 50) |
| Attention Heads | 2 | 2 | – | – | (spatial 2) (temporal 2) (decoder 2) |
| MLP layers (dim) | – | – | 256 | 2 layers (50 nodes) per lag | – |
| Num Variables | 40 | 40 | 40 | 20 | 40 |
| Epochs | 100 | 100 | 100 | 5000 (with early stopping) | 100 |

Table 5: Experiment Configurations for SWaT Benchmark

| Key Parameter | FEDformer ¶ | iTransformer ¶ | CuasalRCA† | AERCA* GVAR** | CrGSTA |
|---|---|---|---|---|---|
| Learning Rate | 1e-4 | 1e-4 | 1e-4 | 1e-6 | 1e-4 |
| Attention Dim | 256 | 256 | – | – | (spatial 256) (temporal 256) (decoder 64) |
| Attention Heads | 2 | 2 | – | – | (spatial 2) (temporal 2) (decoder 2) |
| MLP layers (dim) | – | – | 256 | 8 layers (1000 nodes) per lag | – |
| Epochs | 1000 | 1000 | 1000 | 5000 (with early stopping) | 1000 |

Table 6: Experiment Configurations for NonLinear Benchmark

| Key Parameter | FEDformer ¶ | iTransformer ¶ | CuasalRCA† | AERCA* GVAR** | CrGSTA |
|---|---|---|---|---|---|
| Learning Rate | 1e-4 | 1e-4 | 1e-4 | 1e-4 | 1e-4 |
| Attention Dim | 64 | 64 | – | – | (spatial 64) (temporal 64) (decoder 50) |
| Attention Heads | 2 | 2 | – | – | (spatial 2) (temporal 2) (decoder 2) |
| MLP layers (dim) | – | – | 256 | 8 layers (50 nodes) per lag | – |
| Num Variables | 20 | 20 | 20 | 20 | 20 |
| Epochs | 100 | 100 | 100 | 100 | 100 |

Table 7: Experiment Configurations for MSDS Benchmark

| Key Parameter | FEDformer ¶ | iTransformer ¶ | CuasalRCA† | AERCA* GVAR** | CrGSTA |
|---|---|---|---|---|---|
| Learning Rate | 1e-4 | 1e-4 | 1e-4 | 1e-6 | 1e-4 |
| Attention Dim | 16 | 16 | – | – | (spatial 16) (temporal 16) (decoder 50) |
| Attention Heads | 2 | 2 | – | – | (spatial 2) (temporal 2) (decoder 2) |
| MLP layers (dim) | – | – | 256 | 4 layers (1000 nodes) per lag | – |
| Num Variables | 10 | 10 | 10 | 10 | 10 |
| Epochs | 1000 | 1000 | 1000 | 5000 (with early stopping) | 1000 |

### A.8.1 MOTIVATIONS FOR SPECIFIC PARAMETER CHOICES

**FEDformer ¶, iTransformer ¶**: For FEDformer and iTransformer, for fair comparison, we used the same attention dimension and number of heads as CrGSTA on each dataset. This allows for a more direct comparison of the architectural choices rather than differences in model capacity.

**CausalRCA†**: CausalRCA architecture is relatively simple compared to AERCA and CrGSTA. It is composed of two linear layers with a ReLU activation in between. So for fair comparison, we increased the hidden dimension to 256, to increase the model capacity, to be competitive in performance with other models. This desgin choice allowed CausalRCA to be competitive in performance especially for synthetic datasets, such as Lotka-Volterra and Non-Linear datasets.

**AERCA***: AERCA parameters are adopted from the original implementation Han et al. (2025). We note that AERCA uses a separate MLP per lag to model temporal dependencies, which limits scalability to longer windows. This desgin choice, which has been inherited from GVAR Marcinkevičs & Vogt (2021b), results in a linear increase in the number of parameters as the window size increases. This has been the motivation for our CrGSTA model to use a shared graph structure across time lags, which allows for a more parameter efficient representation.

**GVAR***: GVAR architecture Marcinkevičs & Vogt (2021b) is an encoder that uses a separate MLP per lag to model temporal dependencies. AERCA builds on this by adding two decoders built using the same per-lag MLP structure. Moreover, AERCA added KL regularization, decoder smoothness, and decoder sparsity losses to the training objective. For the reconstruction loss, GVAR reconstructs the whole time series window, while AERCA reconstructs the shifted window due to its encoder-decoder structure. To ensure a fair comparison, we implemented GVAR version using the encoder architecture as AERCA, while removing the decoder modules and their associated losses from the training objective including the KL regularization, and used for reconstruction the whole time series window as in the original GVAR.

### A.9 FULL RESULTS

In this section, we provide the set of full tables and figures for the experiments in RQ1, RQ2 and RQ3 from the main paper. Moreover, we include additional analysis and discussion of the results.

### A.9.1 RQ1 (TEMPORAL DIMENSION) - FULL TABLES

In this experiment, we investigate the impact of varying the temporal window size on root cause identification performance. We evaluate a range of window sizes from 1 to 12 time steps, assessing how this parameter influences the model's ability to accurately identify root causes in both the Lotka–Volterra, Non-Linear SWaT and MSDS datasets.

A.9.1.1 Parameter Scaling.

Figure 8 illustrates how the number of trainable parameters scales with increasing temporal window sizes for all the different datasets. Notably, CrGSTA maintains a relatively stable parameter count across window sizes, demonstrating its efficiency in handling longer temporal contexts without a significant increase in model complexity. In contrast, GVAR and AERCA exhibit a linear growth in parameters as the window size increases, which can lead to scalability challenges for larger windows. This efficiency of CrGSTA is particularly advantageous for practical applications where computational resources may be limited.

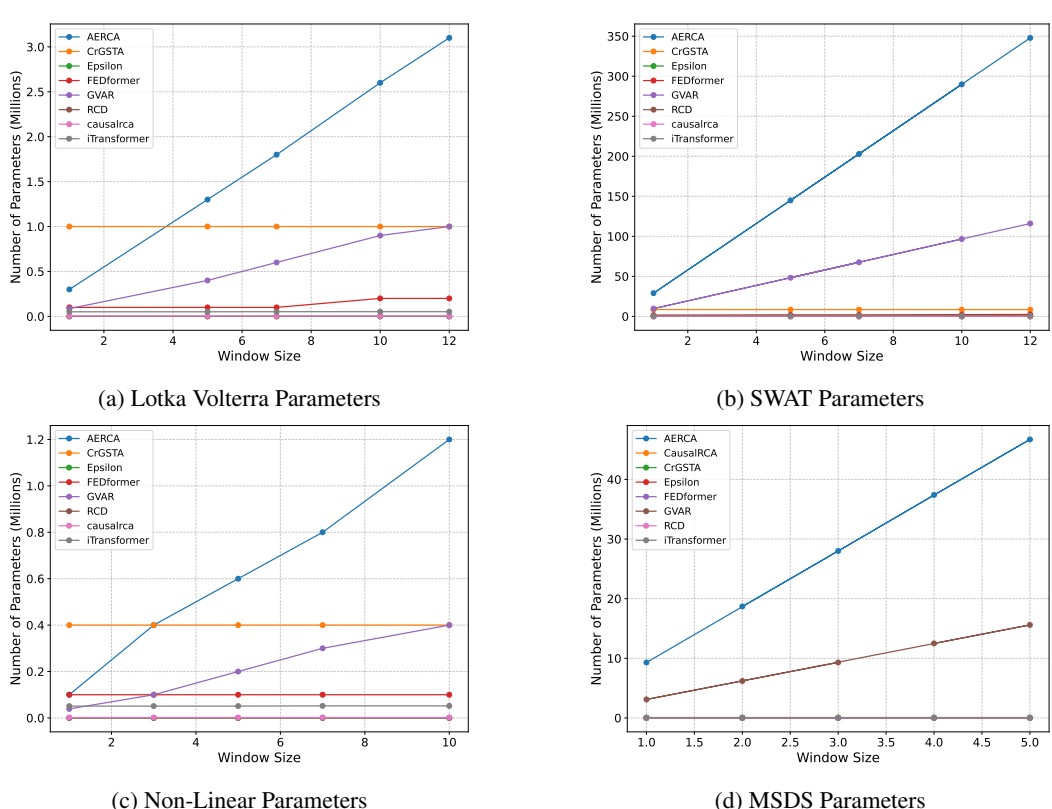

(a) Lotka Volterra Parameters

(b) SWAT Parameters

(c) Non-Linear Parameters

(d) MSDS Parameters

Figure 8: Parameter scaling for Lotka Volterra (left) and SWAT (right) for temporal scaling.

A.9.1.2 Full Results Tables

In this section, we present the complete results for the temporal window size experiments on the different datasets. The tables below summarize the performance of various models across a range of window sizes, providing detailed insights into how temporal context influences root cause identification accuracy. Specifcally, we report the AC@1, AC@3, AC@5, AC@10, and Avg@10 metrics for each model and window size configuration. Moreover, we include the number of trainable parameters for each model to facilitate a comprehensive comparison of performance relative to model complexity. Due to the space limitations in the main paper, we inclue the detailed analysis of both Non-Linear and MSDS datasets here, while the Lotka Volterra results and MSDS are presented in the main paper.

| scheme | Params | window size | AC@1 | AC@3 | AC@5 | AC@10 | Avg@10 |
|---|---|---|---|---|---|---|---|
| | | | LOTKA VOLTERRA | | | | |
| iTransformer | 0.052M | 10 | $0.060_{\pm 0.010}$ | $0.089_{\pm 0.004}$ | $0.120_{\pm 0.005}$ | $0.234_{\pm 0.006}$ | $0.139_{\pm 0.002}$ |
| iTransformer | 0.052M | 7 | $0.090_{\pm 0.017}$ | $0.070_{\pm 0.012}$ | $0.101_{\pm 0.010}$ | $0.222_{\pm 0.018}$ | $0.128_{\pm 0.008}$ |
| FEDformer | 0.2M | 12 | $0.100_{\pm 0.030}$ | $0.099_{\pm 0.004}$ | $0.133_{\pm 0.008}$ | $0.268_{\pm 0.021}$ | $0.162_{\pm 0.012}$ |
| FEDformer | 0.1M | 1 | $0.103_{\pm 0.015}$ | $0.102_{\pm 0.008}$ | $0.115_{\pm 0.010}$ | $0.233_{\pm 0.007}$ | $0.143_{\pm 0.003}$ |
| iTransformer | 0.051M | 5 | $0.107_{\pm 0.032}$ | $0.100_{\pm 0.012}$ | $0.109_{\pm 0.010}$ | $0.234_{\pm 0.014}$ | $0.146_{\pm 0.008}$ |
| iTransformer | 0.052M | 12 | $0.107_{\pm 0.032}$ | $0.090_{\pm 0.006}$ | $0.122_{\pm 0.007}$ | $0.219_{\pm 0.005}$ | $0.140_{\pm 0.002}$ |
| FEDformer | 0.1M | 7 | $0.120_{\pm 0.046}$ | $0.106_{\pm 0.022}$ | $0.129_{\pm 0.016}$ | $0.278_{\pm 0.011}$ | $0.168_{\pm 0.013}$ |
| RCD | 0.000M | 1 | $0.120_{\pm 0.000}$ | $0.150_{\pm 0.000}$ | $0.157_{\pm 0.000}$ | $0.267_{\pm 0.000}$ | $0.185_{\pm 0.000}$ |
| RCD | 0.000M | 5 | $0.120_{\pm 0.000}$ | $0.150_{\pm 0.000}$ | $0.157_{\pm 0.000}$ | $0.267_{\pm 0.000}$ | $0.185_{\pm 0.000}$ |
| RCD | 0.000M | 7 | $0.120_{\pm 0.000}$ | $0.150_{\pm 0.000}$ | $0.157_{\pm 0.000}$ | $0.267_{\pm 0.000}$ | $0.185_{\pm 0.000}$ |
| RCD | 0.000M | 10 | $0.120_{\pm 0.000}$ | $0.150_{\pm 0.000}$ | $0.157_{\pm 0.000}$ | $0.267_{\pm 0.000}$ | $0.185_{\pm 0.000}$ |
| RCD | 0.000M | 12 | $0.120_{\pm 0.000}$ | $0.150_{\pm 0.000}$ | $0.157_{\pm 0.000}$ | $0.267_{\pm 0.000}$ | $0.185_{\pm 0.000}$ |
| iTransformer | 0.051M | 1 | $0.127_{\pm 0.015}$ | $0.112_{\pm 0.013}$ | $0.139_{\pm 0.005}$ | $0.249_{\pm 0.016}$ | $0.166_{\pm 0.012}$ |
| FEDformer | 0.2M | 10 | $0.137_{\pm 0.055}$ | $0.111_{\pm 0.023}$ | $0.132_{\pm 0.014}$ | $0.271_{\pm 0.011}$ | $0.168_{\pm 0.018}$ |
| FEDformer | 0.1M | 5 | $0.140_{\pm 0.036}$ | $0.120_{\pm 0.017}$ | $0.142_{\pm 0.013}$ | $0.275_{\pm 0.028}$ | $0.175_{\pm 0.006}$ |
| Epsilon | 0.000M | 1 | $0.150_{\pm 0.000}$ | $0.113_{\pm 0.000}$ | $0.145_{\pm 0.000}$ | $0.243_{\pm 0.000}$ | $0.167_{\pm 0.000}$ |
| Epsilon | 0.000M | 5 | $0.150_{\pm 0.000}$ | $0.113_{\pm 0.000}$ | $0.145_{\pm 0.000}$ | $0.243_{\pm 0.000}$ | $0.167_{\pm 0.000}$ |
| Epsilon | 0.000M | 7 | $0.150_{\pm 0.000}$ | $0.113_{\pm 0.000}$ | $0.145_{\pm 0.000}$ | $0.243_{\pm 0.000}$ | $0.167_{\pm 0.000}$ |
| Epsilon | 0.000M | 10 | $0.150_{\pm 0.000}$ | $0.113_{\pm 0.000}$ | $0.145_{\pm 0.000}$ | $0.243_{\pm 0.000}$ | $0.167_{\pm 0.000}$ |
| Epsilon | 0.000M | 12 | $0.150_{\pm 0.000}$ | $0.113_{\pm 0.000}$ | $0.145_{\pm 0.000}$ | $0.243_{\pm 0.000}$ | $0.167_{\pm 0.000}$ |
| GVAR | 1.0M | 12 | $0.421_{\pm 0.037}$ | $0.334_{\pm 0.016}$ | $0.362_{\pm 0.018}$ | $0.558_{\pm 0.022}$ | $0.426_{\pm 0.021}$ |
| GVAR | 0.9M | 10 | $0.483_{\pm 0.045}$ | $0.366_{\pm 0.038}$ | $0.390_{\pm 0.026}$ | $0.586_{\pm 0.016}$ | $0.459_{\pm 0.024}$ |
| GVAR | 0.6M | 7 | $0.560_{\pm 0.011}$ | $0.418_{\pm 0.006}$ | $0.436_{\pm 0.019}$ | $0.634_{\pm 0.012}$ | $0.510_{\pm 0.011}$ |
| GVAR | 0.4M | 5 | $0.641_{\pm 0.016}$ | $0.495_{\pm 0.004}$ | $0.514_{\pm 0.009}$ | $0.704_{\pm 0.008}$ | $0.590_{\pm 0.007}$ |
| AERCA | 0.3M | 1 | $0.740_{\pm 0.017}$ | $0.524_{\pm 0.015}$ | $0.488_{\pm 0.018}$ | $0.662_{\pm 0.003}$ | $0.584_{\pm 0.010}$ |
| CrGSTA | 1.0M | 1 | $0.750_{\pm 0.026}$ | $0.520_{\pm 0.019}$ | $0.481_{\pm 0.023}$ | $0.648_{\pm 0.007}$ | $0.576_{\pm 0.014}$ |
| GVAR | 0.086M | 1 | $0.763_{\pm 0.016}$ | $0.547_{\pm 0.011}$ | $0.531_{\pm 0.009}$ | $0.705_{\pm 0.007}$ | $0.621_{\pm 0.008}$ |
| CrGSTA | 1.0M | 5 | $0.770_{\pm 0.030}$ | $0.524_{\pm 0.032}$ | $0.493_{\pm 0.010}$ | $0.658_{\pm 0.004}$ | $0.590_{\pm 0.012}$ |
| CrGSTA | 1.0M | 12 | $0.770_{\pm 0.046}$ | $0.513_{\pm 0.018}$ | $0.486_{\pm 0.012}$ | $0.661_{\pm 0.013}$ | $0.585_{\pm 0.017}$ |
| CrGSTA | 1.0M | 10 | $0.880_{\pm 0.028}$ | $0.663_{\pm 0.009}$ | $0.589_{\pm 0.014}$ | $0.748_{\pm 0.005}$ | $0.694_{\pm 0.003}$ |
| CrGSTA | 1.0M | 7 | $0.930_{\pm 0.028}$ | $0.753_{\pm 0.000}$ | $0.682_{\pm 0.011}$ | $\mathbf{0.845_{\pm 0.004}}$ | $0.782_{\pm 0.008}$ |
| AERCA | 3.1M | 12 | $0.930_{\pm 0.014}$ | $0.703_{\pm 0.014}$ | $0.666_{\pm 0.004}$ | $0.805_{\pm 0.010}$ | $0.758_{\pm 0.003}$ |
| AERCA | 2.6M | 10 | $0.935_{\pm 0.007}$ | $0.735_{\pm 0.012}$ | $0.669_{\pm 0.022}$ | $\underline{0.817_{\pm 0.006}}$ | $0.769_{\pm 0.007}$ |
| causalrca | 0.004M | 1 | $0.963_{\pm 0.004}$ | $0.742_{\pm 0.011}$ | $0.642_{\pm 0.007}$ | $0.763_{\pm 0.002}$ | $0.745_{\pm 0.005}$ |
| causalrca | 0.004M | 5 | $0.963_{\pm 0.004}$ | $0.741_{\pm 0.011}$ | $0.642_{\pm 0.007}$ | $0.762_{\pm 0.003}$ | $0.745_{\pm 0.005}$ |
| causalrca | 0.004M | 7 | $0.963_{\pm 0.004}$ | $0.742_{\pm 0.011}$ | $0.641_{\pm 0.007}$ | $0.762_{\pm 0.003}$ | $0.745_{\pm 0.006}$ |
| causalrca | 0.004M | 10 | $0.963_{\pm 0.004}$ | $0.742_{\pm 0.011}$ | $0.641_{\pm 0.007}$ | $0.762_{\pm 0.002}$ | $0.745_{\pm 0.005}$ |
| causalrca | 0.004M | 12 | $0.963_{\pm 0.004}$ | $0.742_{\pm 0.011}$ | $0.641_{\pm 0.007}$ | $0.762_{\pm 0.003}$ | $0.745_{\pm 0.005}$ |
| AERCA | 1.8M | 7 | $\underline{0.970_{\pm 0.026}}$ | $\underline{0.764_{\pm 0.031}}$ | $\underline{0.697_{\pm 0.023}}$ | $0.814_{\pm 0.026}$ | $\underline{0.791_{\pm 0.017}}$ |
| AERCA | 1.3M | 5 | $\mathbf{0.977_{\pm 0.006}}$ | $\mathbf{0.788_{\pm 0.007}}$ | $\mathbf{0.717_{\pm 0.010}}$ | $0.816_{\pm 0.008}$ | $\mathbf{0.803_{\pm 0.003}}$ |

Table 8: RQ1 Lotka Windows

| scheme | Params | window size | AC@1 | AC@3 | AC@5 | AC@10 | Avg@10 |
|---|---|---|---|---|---|---|---|
| | | | | SWAT | | | |
| GVAR | 116.0M | 12 | - | - | - | - | - |
| AERCA | 144.9M | 5 | - | - | - | - | - |
| AERCA | 202.9M | 7 | - | - | - | - | - |
| AERCA | 289.9M | 10 | - | - | - | - | - |
| AERCA | 347.9M | 12 | - | - | - | - | - |
| Epsilon | 0.000M | 5 | $0.000_{\pm0.000}$ | $0.100_{\pm0.000}$ | $0.100_{\pm0.000}$ | $0.300_{\pm0.000}$ | $0.140_{\pm0.000}$ |
| Epsilon | 0.000M | 12 | $0.000_{\pm0.000}$ | $0.025_{\pm0.000}$ | $0.025_{\pm0.000}$ | $0.275_{\pm0.000}$ | $0.070_{\pm0.000}$ |
| RCD | 0.000M | 1 | $0.000_{\pm0.000}$ | $0.000_{\pm0.000}$ | $0.000_{\pm0.000}$ | $0.300_{\pm0.000}$ | $0.100_{\pm0.000}$ |
| RCD | 0.000M | 5 | $0.000_{\pm0.000}$ | $0.000_{\pm0.000}$ | $0.000_{\pm0.000}$ | $0.300_{\pm0.000}$ | $0.100_{\pm0.000}$ |
| RCD | 0.000M | 7 | $0.000_{\pm0.000}$ | $0.000_{\pm0.000}$ | $0.000_{\pm0.000}$ | $0.300_{\pm0.000}$ | $0.100_{\pm0.000}$ |
| RCD | 0.000M | 10 | $0.000_{\pm0.000}$ | $0.000_{\pm0.000}$ | $0.000_{\pm0.000}$ | $0.300_{\pm0.000}$ | $0.100_{\pm0.000}$ |
| RCD | 0.000M | 12 | $0.000_{\pm0.000}$ | $0.000_{\pm0.000}$ | $0.000_{\pm0.000}$ | $0.300_{\pm0.000}$ | $0.100_{\pm0.000}$ |
| iTransformer | 0.8M | 10 | $0.031_{\pm0.002}$ | $0.116_{\pm0.008}$ | $0.215_{\pm0.008}$ | $0.387_{\pm0.014}$ | $0.208_{\pm0.002}$ |
| iTransformer | 0.8M | 7 | $0.043_{\pm0.000}$ | $0.108_{\pm0.009}$ | $0.170_{\pm0.013}$ | $0.417_{\pm0.015}$ | $0.206_{\pm0.004}$ |
| iTransformer | 0.8M | 12 | $0.044_{\pm0.002}$ | $0.098_{\pm0.007}$ | $0.215_{\pm0.005}$ | $0.364_{\pm0.006}$ | $0.203_{\pm0.002}$ |
| Epsilon | 0.000M | 7 | $0.050_{\pm0.000}$ | $0.075_{\pm0.000}$ | $0.075_{\pm0.000}$ | $0.375_{\pm0.000}$ | $0.110_{\pm0.000}$ |
| Epsilon | 0.000M | 10 | $0.050_{\pm0.000}$ | $0.100_{\pm0.000}$ | $0.100_{\pm0.000}$ | $0.300_{\pm0.000}$ | $0.142_{\pm0.000}$ |
| FEDformer | 2.4M | 12 | $0.054_{\pm0.000}$ | $0.058_{\pm0.000}$ | $0.109_{\pm0.003}$ | $0.235_{\pm0.040}$ | $0.124_{\pm0.008}$ |
| causalrca | 0.007M | 7 | $0.058_{\pm0.029}$ | $0.084_{\pm0.035}$ | $0.122_{\pm0.031}$ | $0.261_{\pm0.007}$ | $0.148_{\pm0.022}$ |
| FEDformer | 2.2M | 10 | $0.060_{\pm0.000}$ | $0.061_{\pm0.001}$ | $0.113_{\pm0.000}$ | $0.278_{\pm0.010}$ | $0.134_{\pm0.003}$ |
| FEDformer | 1.9M | 7 | $0.064_{\pm0.000}$ | $0.068_{\pm0.000}$ | $0.115_{\pm0.006}$ | $0.292_{\pm0.029}$ | $0.142_{\pm0.003}$ |
| causalrca | 0.007M | 5 | $0.069_{\pm0.037}$ | $0.109_{\pm0.040}$ | $0.127_{\pm0.038}$ | $0.357_{\pm0.029}$ | $0.178_{\pm0.031}$ |
| FEDformer | 1.9M | 5 | $0.070_{\pm0.000}$ | $0.083_{\pm0.004}$ | $0.133_{\pm0.004}$ | $0.297_{\pm0.028}$ | $0.152_{\pm0.005}$ |
| iTransformer | 0.8M | 5 | $0.073_{\pm0.010}$ | $0.143_{\pm0.004}$ | $0.250_{\pm0.012}$ | $0.498_{\pm0.013}$ | $0.267_{\pm0.004}$ |
| Epsilon | 0.000M | 1 | $0.100_{\pm0.000}$ | $0.150_{\pm0.000}$ | $0.150_{\pm0.000}$ | $0.350_{\pm0.000}$ | $0.170_{\pm0.000}$ |
| causalrca | 0.007M | 1 | $0.117_{\pm0.035}$ | $0.164_{\pm0.036}$ | $0.197_{\pm0.045}$ | $0.294_{\pm0.027}$ | $0.214_{\pm0.031}$ |
| AERCA | 29.0M | 1 | $0.150_{\pm0.045}$ | $0.250_{\pm0.045}$ | $0.317_{\pm0.026}$ | $0.342_{\pm0.038}$ | $0.289_{\pm0.004}$ |
| iTransformer | 0.8M | 1 | $0.150_{\pm0.060}$ | $0.217_{\pm0.067}$ | $0.279_{\pm0.099}$ | $0.400_{\pm0.117}$ | $0.285_{\pm0.083}$ |
| GVAR | 9.7M | 1 | $0.167_{\pm0.021}$ | $0.282_{\pm0.059}$ | $0.341_{\pm0.054}$ | $0.475_{\pm0.098}$ | $0.348_{\pm0.052}$ |
| causalrca | 0.007M | 10 | $0.185_{\pm0.017}$ | $0.239_{\pm0.003}$ | $0.248_{\pm0.014}$ | $0.351_{\pm0.021}$ | $0.263_{\pm0.011}$ |
| FEDformer | 1.6M | 1 | $0.208_{\pm0.019}$ | $0.325_{\pm0.000}$ | $0.325_{\pm0.000}$ | $0.498_{\pm0.039}$ | $0.349_{\pm0.008}$ |
| causalrca | 0.007M | 12 | $0.209_{\pm0.005}$ | $0.237_{\pm0.000}$ | $0.242_{\pm0.000}$ | $0.313_{\pm0.011}$ | $0.251_{\pm0.002}$ |
| CrGSTA | 8.5M | 1 | $0.225_{\pm0.104}$ | $0.275_{\pm0.061}$ | $0.333_{\pm0.041}$ | $0.417_{\pm0.054}$ | $0.327_{\pm0.045}$ |
| CrGSTA | 8.5M | 12 | $0.275_{\pm0.028}$ | $0.369_{\pm0.036}$ | $0.415_{\pm0.052}$ | $0.469_{\pm0.058}$ | $0.404_{\pm0.046}$ |
| GVAR | 48.3M | 5 | $0.277_{\pm0.024}$ | $0.388_{\pm0.024}$ | $0.419_{\pm0.039}$ | $0.521_{\pm0.051}$ | $0.429_{\pm0.026}$ |
| CrGSTA | 8.5M | 5 | $0.282_{\pm0.056}$ | $0.361_{\pm0.035}$ | $0.366_{\pm0.035}$ | $0.412_{\pm0.078}$ | $0.368_{\pm0.042}$ |
| GVAR | 67.6M | 7 | $0.285_{\pm0.020}$ | $\underline{0.411_{\pm0.028}}$ | $\underline{0.458_{\pm0.023}}$ | $\underline{0.564_{\pm0.038}}$ | $\underline{0.462_{\pm0.027}}$ |
| CrGSTA | 8.5M | 10 | $0.305_{\pm0.047}$ | $0.397_{\pm0.065}$ | $0.428_{\pm0.065}$ | $0.472_{\pm0.056}$ | $0.418_{\pm0.057}$ |
| CrGSTA | 8.5M | 7 | $0.308_{\pm0.037}$ | $0.396_{\pm0.034}$ | $0.433_{\pm0.030}$ | $0.488_{\pm0.054}$ | $0.426_{\pm0.031}$ |
| GVAR | 96.6M | 10 | $\mathbf{0.339_{\pm0.004}}$ | $\mathbf{0.462_{\pm0.019}}$ | $\mathbf{0.502_{\pm0.037}}$ | $\mathbf{0.592_{\pm0.039}}$ | $\mathbf{0.502_{\pm0.025}}$ |

Table 9: RQ1 Swat Windows

| scheme | Params | window size | AC@1 | AC@3 | AC@5 | AC@10 | Avg@10 |
|--------|--------|-------------|------|------|------|-------|--------|
| | | | | NONLINEAR | | | |
| Epsilon | 0.000M | 1 | $0.180_{\pm 0.000}$ | $0.307_{\pm 0.000}$ | $0.341_{\pm 0.000}$ | $0.530_{\pm 0.000}$ | $0.359_{\pm 0.000}$ |
| Epsilon | 0.000M | 3 | $0.180_{\pm 0.000}$ | $0.307_{\pm 0.000}$ | $0.341_{\pm 0.000}$ | $0.530_{\pm 0.000}$ | $0.359_{\pm 0.000}$ |
| Epsilon | 0.000M | 5 | $0.180_{\pm 0.000}$ | $0.307_{\pm 0.000}$ | $0.341_{\pm 0.000}$ | $0.530_{\pm 0.000}$ | $0.359_{\pm 0.000}$ |
| Epsilon | 0.000M | 7 | $0.180_{\pm 0.000}$ | $0.307_{\pm 0.000}$ | $0.341_{\pm 0.000}$ | $0.530_{\pm 0.000}$ | $0.359_{\pm 0.000}$ |
| Epsilon | 0.000M | 10 | $0.180_{\pm 0.000}$ | $0.307_{\pm 0.000}$ | $0.341_{\pm 0.000}$ | $0.530_{\pm 0.000}$ | $0.359_{\pm 0.000}$ |
| FEDformer | 0.1M | 5 | $0.217_{\pm 0.018}$ | $0.262_{\pm 0.013}$ | $0.281_{\pm 0.010}$ | $0.429_{\pm 0.012}$ | $0.306_{\pm 0.009}$ |
| iTransformer | 0.051M | 5 | $0.225_{\pm 0.012}$ | $0.262_{\pm 0.006}$ | $0.304_{\pm 0.005}$ | $0.453_{\pm 0.006}$ | $0.322_{\pm 0.002}$ |
| FEDformer | 0.1M | 10 | $0.230_{\pm 0.031}$ | $0.255_{\pm 0.018}$ | $0.294_{\pm 0.016}$ | $0.480_{\pm 0.013}$ | $0.328_{\pm 0.011}$ |
| RCD | 0.000M | 1 | $0.230_{\pm 0.000}$ | $0.250_{\pm 0.000}$ | $0.286_{\pm 0.000}$ | $0.519_{\pm 0.000}$ | $0.337_{\pm 0.000}$ |
| RCD | 0.000M | 3 | $0.230_{\pm 0.000}$ | $0.250_{\pm 0.000}$ | $0.286_{\pm 0.000}$ | $0.519_{\pm 0.000}$ | $0.337_{\pm 0.000}$ |
| RCD | 0.000M | 5 | $0.230_{\pm 0.000}$ | $0.250_{\pm 0.000}$ | $0.286_{\pm 0.000}$ | $0.519_{\pm 0.000}$ | $0.337_{\pm 0.000}$ |
| RCD | 0.000M | 7 | $0.230_{\pm 0.000}$ | $0.250_{\pm 0.000}$ | $0.286_{\pm 0.000}$ | $0.519_{\pm 0.000}$ | $0.337_{\pm 0.000}$ |
| RCD | 0.000M | 10 | $0.230_{\pm 0.000}$ | $0.250_{\pm 0.000}$ | $0.286_{\pm 0.000}$ | $0.519_{\pm 0.000}$ | $0.337_{\pm 0.000}$ |
| FEDformer | 0.1M | 3 | $0.235_{\pm 0.013}$ | $0.312_{\pm 0.006}$ | $0.365_{\pm 0.005}$ | $0.536_{\pm 0.011}$ | $0.387_{\pm 0.005}$ |
| FEDformer | 0.1M | 7 | $0.250_{\pm 0.024}$ | $0.257_{\pm 0.016}$ | $0.292_{\pm 0.011}$ | $0.469_{\pm 0.015}$ | $0.324_{\pm 0.005}$ |
| iTransformer | 0.052M | 7 | $0.251_{\pm 0.026}$ | $0.245_{\pm 0.019}$ | $0.282_{\pm 0.019}$ | $0.463_{\pm 0.017}$ | $0.319_{\pm 0.014}$ |
| iTransformer | 0.051M | 1 | $0.253_{\pm 0.057}$ | $0.280_{\pm 0.039}$ | $0.326_{\pm 0.020}$ | $0.507_{\pm 0.028}$ | $0.357_{\pm 0.028}$ |
| iTransformer | 0.052M | 10 | $0.284_{\pm 0.028}$ | $0.277_{\pm 0.013}$ | $0.302_{\pm 0.013}$ | $0.462_{\pm 0.014}$ | $0.337_{\pm 0.008}$ |
| iTransformer | 0.051M | 3 | $0.307_{\pm 0.012}$ | $0.311_{\pm 0.006}$ | $0.357_{\pm 0.006}$ | $0.514_{\pm 0.004}$ | $0.383_{\pm 0.002}$ |
| FEDformer | 0.1M | 1 | $0.317_{\pm 0.009}$ | $0.285_{\pm 0.004}$ | $0.305_{\pm 0.005}$ | $0.497_{\pm 0.005}$ | $0.352_{\pm 0.001}$ |
| AERCA | 1.2M | 10 | $0.322_{\pm 0.057}$ | $0.348_{\pm 0.039}$ | $0.388_{\pm 0.027}$ | $0.565_{\pm 0.022}$ | $0.416_{\pm 0.027}$ |
| AERCA | 0.4M | 3 | $0.337_{\pm 0.064}$ | $0.364_{\pm 0.035}$ | $0.391_{\pm 0.035}$ | $0.565_{\pm 0.044}$ | $0.424_{\pm 0.039}$ |
| AERCA | 0.6M | 5 | $0.337_{\pm 0.072}$ | $0.349_{\pm 0.040}$ | $0.393_{\pm 0.043}$ | $0.568_{\pm 0.036}$ | $0.423_{\pm 0.043}$ |
| AERCA | 0.8M | 7 | $0.341_{\pm 0.043}$ | $0.342_{\pm 0.040}$ | $0.382_{\pm 0.040}$ | $0.557_{\pm 0.038}$ | $0.416_{\pm 0.036}$ |
| GVAR | 0.4M | 10 | $0.360_{\pm 0.040}$ | $0.376_{\pm 0.036}$ | $0.406_{\pm 0.027}$ | $0.569_{\pm 0.024}$ | $0.434_{\pm 0.026}$ |
| GVAR | 0.2M | 5 | $0.368_{\pm 0.038}$ | $0.366_{\pm 0.041}$ | $0.383_{\pm 0.033}$ | $0.537_{\pm 0.032}$ | $0.417_{\pm 0.033}$ |
| GVAR | 0.1M | 3 | $0.387_{\pm 0.071}$ | $0.387_{\pm 0.047}$ | $0.412_{\pm 0.042}$ | $0.570_{\pm 0.044}$ | $0.444_{\pm 0.047}$ |
| AERCA | 0.1M | 1 | $0.394_{\pm 0.034}$ | $0.381_{\pm 0.036}$ | $0.429_{\pm 0.039}$ | $0.606_{\pm 0.030}$ | $0.464_{\pm 0.030}$ |
| GVAR | 0.3M | 7 | $0.395_{\pm 0.058}$ | $0.377_{\pm 0.035}$ | $0.406_{\pm 0.025}$ | $0.557_{\pm 0.030}$ | $0.434_{\pm 0.026}$ |
| CrGSTA | 0.4M | 1 | $0.444_{\pm 0.031}$ | $0.433_{\pm 0.026}$ | $0.463_{\pm 0.016}$ | $0.609_{\pm 0.016}$ | $0.488_{\pm 0.016}$ |
| CrGSTA | 0.4M | 10 | $0.467_{\pm 0.059}$ | $0.459_{\pm 0.029}$ | $0.470_{\pm 0.018}$ | $0.623_{\pm 0.016}$ | $0.504_{\pm 0.015}$ |
| CrGSTA | 0.4M | 7 | $0.473_{\pm 0.024}$ | $0.468_{\pm 0.024}$ | $0.476_{\pm 0.015}$ | $0.614_{\pm 0.022}$ | $0.508_{\pm 0.013}$ |
| GVAR | 0.039M | 1 | $0.484_{\pm 0.047}$ | $0.448_{\pm 0.049}$ | $0.464_{\pm 0.044}$ | $0.605_{\pm 0.045}$ | $0.498_{\pm 0.043}$ |
| CrGSTA | 0.4M | 5 | $0.490_{\pm 0.031}$ | $\underline{0.477_{\pm 0.019}}$ | $\underline{0.487_{\pm 0.022}}$ | $\underline{0.626_{\pm 0.014}}$ | $\underline{0.517_{\pm 0.013}}$ |
| CrGSTA | 0.4M | 3 | $0.492_{\pm 0.046}$ | $\mathbf{0.489_{\pm 0.015}}$ | $\mathbf{0.499_{\pm 0.022}}$ | $\mathbf{0.641_{\pm 0.025}}$ | $\mathbf{0.528_{\pm 0.016}}$ |
| causalrca | 0.002M | 5 | $0.500_{\pm 0.031}$ | $0.477_{\pm 0.028}$ | $0.474_{\pm 0.027}$ | $0.608_{\pm 0.021}$ | $0.515_{\pm 0.025}$ |
| causalrca | 0.002M | 7 | $0.500_{\pm 0.029}$ | $0.475_{\pm 0.027}$ | $0.473_{\pm 0.028}$ | $0.610_{\pm 0.021}$ | $0.515_{\pm 0.025}$ |
| causalrca | 0.002M | 10 | $0.501_{\pm 0.030}$ | $0.475_{\pm 0.027}$ | $0.473_{\pm 0.028}$ | $0.609_{\pm 0.021}$ | $0.515_{\pm 0.025}$ |
| causalrca | 0.002M | 1 | $\underline{0.503_{\pm 0.030}}$ | $0.477_{\pm 0.026}$ | $0.475_{\pm 0.027}$ | $0.608_{\pm 0.020}$ | $0.516_{\pm 0.025}$ |
| causalrca | 0.002M | 3 | $\mathbf{0.503_{\pm 0.033}}$ | $0.475_{\pm 0.026}$ | $0.474_{\pm 0.027}$ | $0.608_{\pm 0.020}$ | $0.515_{\pm 0.025}$ |

Table 10: RQ1 Non-Linear Windows

**Non-Linear Dataset Results.** Similar trends are observed in the Non-Linear dataset (Table 10) as in the synthetic Lotka–Volterra dataset. Statistical methods remain largely flat (Avg@10 $\approx$ 0.36–0.37), reflecting their inability to capture nonlinear dependencies. Non-causal deep models exhibit mild temporal sensitivity but saturate quickly: FEDformer peaks at window 1 (0.352), while iTransformer reaches 0.383 at window 3. Simple causal models, such as causalrca with 256-unit MLPs, are capable of modeling relatively complex dependencies but plateau at 0.516 Avg@10. This limitation may be attributed to the characteristics of the synthetic Non-Linear dataset, which restricts the exploration of longer-term temporal dependencies compared to real-world datasets like SWaT. GVAR, constrained by its encoder-only architecture, fails to fully exploit longer temporal windows, and its performance even shows a slight decline as the window size increases. AERCA similarly exhibits reduced performance with longer windows, likely due to overfitting given its substantially larger parameter count. In contrast, CrGSTA achieves the best accuracy under a fixed parameter budget

(0.528 at window 3), with improvements driven primarily by cross-domain temporal modeling rather than model size.

| scheme | Params | window size | AC@1 | AC@3 | AC@5 | AC@10 | Avg@10 |
|--------|--------|-------------|------|------|------|-------|--------|
| | | | | MSDS | | | |
| iTransformer | 0.004M | 2 | $0.000_{\pm0.000}$ | $0.043_{\pm0.000}$ | $0.283_{\pm0.000}$ | $1.000_{\pm0.000}$ | $0.399_{\pm0.006}$ |
| iTransformer | 0.004M | 3 | $0.000_{\pm0.000}$ | $0.000_{\pm0.000}$ | $0.138_{\pm0.017}$ | $1.000_{\pm0.000}$ | $0.364_{\pm0.010}$ |
| iTransformer | 0.004M | 4 | $0.000_{\pm0.000}$ | $0.000_{\pm0.000}$ | $0.149_{\pm0.029}$ | $1.000_{\pm0.000}$ | $0.362_{\pm0.009}$ |
| iTransformer | 0.004M | 5 | $0.000_{\pm0.000}$ | $0.000_{\pm0.000}$ | $0.031_{\pm0.024}$ | $1.000_{\pm0.000}$ | $0.331_{\pm0.016}$ |
| CausalRCA | 0.002M | 1 | $0.109_{\pm0.000}$ | $0.355_{\pm0.018}$ | $0.652_{\pm0.000}$ | $1.000_{\pm0.000}$ | $0.629_{\pm0.003}$ |
| CausalRCA | 0.002M | 2 | $0.109_{\pm0.000}$ | $0.362_{\pm0.026}$ | $0.652_{\pm0.000}$ | $1.000_{\pm0.000}$ | $0.628_{\pm0.006}$ |
| CausalRCA | 0.002M | 3 | $0.109_{\pm0.000}$ | $0.351_{\pm0.021}$ | $0.652_{\pm0.000}$ | $1.000_{\pm0.000}$ | $0.628_{\pm0.005}$ |
| CausalRCA | 0.002M | 4 | $0.109_{\pm0.000}$ | $0.362_{\pm0.018}$ | $0.652_{\pm0.000}$ | $1.000_{\pm0.000}$ | $0.627_{\pm0.006}$ |
| CausalRCA | 0.002M | 5 | $0.109_{\pm0.000}$ | $0.362_{\pm0.033}$ | $0.652_{\pm0.000}$ | $1.000_{\pm0.000}$ | $0.629_{\pm0.004}$ |
| GVAR | 6.2M | 2 | $0.203_{\pm0.207}$ | $0.917_{\pm0.021}$ | $1.000_{\pm0.000}$ | $1.000_{\pm0.000}$ | $0.850_{\pm0.022}$ |
| GVAR | 9.3M | 3 | $0.283_{\pm0.151}$ | $0.935_{\pm0.024}$ | $1.000_{\pm0.000}$ | $1.000_{\pm0.000}$ | $0.859_{\pm0.019}$ |
| Epsilon | 0.000M | 1 | $0.283_{\pm0.000}$ | $0.304_{\pm0.000}$ | $0.826_{\pm0.000}$ | $1.000_{\pm0.000}$ | $0.739_{\pm0.000}$ |
| Epsilon | 0.000M | 2 | $0.283_{\pm0.000}$ | $0.304_{\pm0.000}$ | $0.826_{\pm0.000}$ | $1.000_{\pm0.000}$ | $0.739_{\pm0.000}$ |
| Epsilon | 0.000M | 3 | $0.283_{\pm0.000}$ | $0.304_{\pm0.000}$ | $0.826_{\pm0.000}$ | $1.000_{\pm0.000}$ | $0.739_{\pm0.000}$ |
| Epsilon | 0.000M | 4 | $0.283_{\pm0.000}$ | $0.304_{\pm0.000}$ | $0.826_{\pm0.000}$ | $1.000_{\pm0.000}$ | $0.739_{\pm0.000}$ |
| Epsilon | 0.000M | 5 | $0.283_{\pm0.000}$ | $0.304_{\pm0.000}$ | $0.826_{\pm0.000}$ | $1.000_{\pm0.000}$ | $0.739_{\pm0.000}$ |
| iTransformer | 0.003M | 1 | $0.301_{\pm0.279}$ | $0.426_{\pm0.400}$ | $0.486_{\pm0.401}$ | $1.000_{\pm0.000}$ | $0.627_{\pm0.222}$ |
| AERCA | 9.3M | 1 | $0.330_{\pm0.093}$ | $0.946_{\pm0.051}$ | $1.000_{\pm0.000}$ | $1.000_{\pm0.000}$ | $0.868_{\pm0.020}$ |
| GVAR | 12.5M | 4 | $0.351_{\pm0.098}$ | $0.913_{\pm0.014}$ | $\mathbf{1.000_{\pm0.000}}$ | $1.000_{\pm0.000}$ | $0.865_{\pm0.012}$ |
| GVAR | 15.6M | 5 | $0.370_{\pm0.024}$ | $0.880_{\pm0.045}$ | $\underline{1.000_{\pm0.000}}$ | $1.000_{\pm0.000}$ | $0.861_{\pm0.007}$ |
| GVAR | 3.1M | 1 | $0.377_{\pm0.022}$ | $0.971_{\pm0.061}$ | $\underline{1.000_{\pm0.000}}$ | $1.000_{\pm0.000}$ | $0.893_{\pm0.026}$ |
| FEDformer | 0.008M | 1 | $0.386_{\pm0.019}$ | $0.442_{\pm0.176}$ | $0.730_{\pm0.265}$ | $\mathbf{1.000_{\pm0.000}}$ | $0.723_{\pm0.100}$ |
| CrGSTA | 0.069M | 5 | $0.386_{\pm0.094}$ | $0.859_{\pm0.190}$ | $0.981_{\pm0.039}$ | $1.000_{\pm0.000}$ | $0.877_{\pm0.058}$ |
| FEDformer | 0.008M | 2 | $0.391_{\pm0.000}$ | $0.399_{\pm0.025}$ | $0.828_{\pm0.243}$ | $\underline{1.000_{\pm0.000}}$ | $0.757_{\pm0.058}$ |
| FEDformer | 0.008M | 3 | $0.391_{\pm0.000}$ | $0.399_{\pm0.025}$ | $0.832_{\pm0.239}$ | $1.000_{\pm0.000}$ | $0.756_{\pm0.057}$ |
| FEDformer | 0.009M | 4 | $0.391_{\pm0.000}$ | $0.498_{\pm0.235}$ | $0.895_{\pm0.236}$ | $1.000_{\pm0.000}$ | $0.789_{\pm0.067}$ |
| FEDformer | 0.009M | 5 | $0.391_{\pm0.000}$ | $0.498_{\pm0.235}$ | $0.899_{\pm0.237}$ | $1.000_{\pm0.000}$ | $0.790_{\pm0.066}$ |
| AERCA | 18.7M | 2 | $0.391_{\pm nan}$ | $0.674_{\pm nan}$ | $1.000_{\pm nan}$ | $1.000_{\pm nan}$ | $0.841_{\pm nan}$ |
| AERCA | 28.0M | 3 | $0.391_{\pm0.000}$ | $0.884_{\pm0.123}$ | $1.000_{\pm0.000}$ | $1.000_{\pm0.000}$ | $0.864_{\pm0.016}$ |
| AERCA | 37.4M | 4 | $0.391_{\pm0.000}$ | $0.848_{\pm0.235}$ | $0.978_{\pm0.053}$ | $1.000_{\pm0.000}$ | $0.864_{\pm0.046}$ |
| AERCA | 46.7M | 5 | $0.391_{\pm0.000}$ | $0.746_{\pm0.221}$ | $0.975_{\pm0.062}$ | $1.000_{\pm0.000}$ | $0.840_{\pm0.053}$ |
| CrGSTA | 0.069M | 1 | $0.435_{\pm0.100}$ | $0.949_{\pm0.081}$ | $0.998_{\pm0.007}$ | $1.000_{\pm0.000}$ | $0.921_{\pm0.025}$ |
| CrGSTA | 0.069M | 2 | $0.498_{\pm0.133}$ | $0.971_{\pm0.049}$ | $1.000_{\pm0.000}$ | $1.000_{\pm0.000}$ | $\underline{0.933_{\pm0.036}}$ |
| CrGSTA | 0.069M | 4 | $0.498_{\pm0.098}$ | $0.879_{\pm0.195}$ | $0.969_{\pm0.094}$ | $1.000_{\pm0.000}$ | $0.902_{\pm0.066}$ |
| CrGSTA | 0.069M | 3 | $0.543_{\pm0.082}$ | $0.971_{\pm0.049}$ | $1.000_{\pm0.000}$ | $1.000_{\pm0.000}$ | $\mathbf{0.937_{\pm0.030}}$ |
| RCD | 0.000M | 1 | $\mathbf{0.609_{\pm0.000}}$ | $\mathbf{1.000_{\pm0.000}}$ | $1.000_{\pm0.000}$ | $1.000_{\pm0.000}$ | $0.922_{\pm0.000}$ |
| RCD | 0.000M | 2 | $\underline{0.609_{\pm0.000}}$ | $\underline{1.000_{\pm0.000}}$ | $1.000_{\pm0.000}$ | $1.000_{\pm0.000}$ | $0.922_{\pm0.000}$ |
| RCD | 0.000M | 3 | $0.609_{\pm0.000}$ | $1.000_{\pm0.000}$ | $1.000_{\pm0.000}$ | $1.000_{\pm0.000}$ | $0.922_{\pm0.000}$ |
| RCD | 0.000M | 4 | $0.609_{\pm0.000}$ | $1.000_{\pm0.000}$ | $1.000_{\pm0.000}$ | $1.000_{\pm0.000}$ | $0.922_{\pm0.000}$ |
| RCD | 0.000M | 5 | $0.609_{\pm0.000}$ | $1.000_{\pm0.000}$ | $1.000_{\pm0.000}$ | $1.000_{\pm0.000}$ | $0.922_{\pm0.000}$ |

Table 11: RQ1 MSDS Windows

**MSDS Dataset Results.** Results on the MSDS dataset (Table 11) further validate our observations. MSDS represents a real-world cloud computing environment with a relatively small number of variables (10), allowing models to fully exploit temporal dependencies. Statistical methods (Epsilon, RCD) perform consistently but are outperformed by deep models. Notably, RCD achieves high accuracy (0.922 Avg@10) despite its simplicity, likely due to the limited variable count of MSDS, but its performance remains unchanged across window sizes. Among deep models, iTransformer shows limited temporal sensitivity, while FEDformer, likely due to its frequency-domain design, responds more to changes in window size. Causal models such as causalrca plateau quickly (0.629 Avg@10), reflecting their simpler architecture. GVAR achieves stronger performance (0.893 Avg@10) by

effectively modeling temporal dependencies but does not improve with larger windows. AERCA shows a similar trend, with performance declining slightly for longer windows, likely due to overfitting from its larger parameter count. In contrast, CrGSTA consistently attains the best performance across all window sizes, peaking at 0.937 Avg@10 at window 3, demonstrating its ability to capture temporal dependencies efficiently in real-world settings.

### A.9.2  RQ2 (SPATIAL DIMENSION) – FULL TABLE

In this experiment, we evaluate how varying the number of variables (spatial dimension) affects root cause identification performance. We test variable counts from 20 to 60 on the Lotka–Volterra dataset and from 15 to 35 on Non-Linear dataset, maintaining a fixed temporal window size for all models.

#### A.9.2.1  Lotka–Volterra Dataset Results.

| scheme | Params | num vars | AC@1 | AC@3 | AC@5 | AC@10 | Avg@10 |
|---|---|---|---|---|---|---|---|
| | | | LOTKA VOLTERRA | | | | |
| FEDformer | 0.1M | 50 | $0.073_{\pm 0.015}$ | $0.077_{\pm 0.007}$ | $0.097_{\pm 0.003}$ | $0.191_{\pm 0.010}$ | $0.118_{\pm 0.001}$ |
| FEDformer | 0.2M | 60 | $0.077_{\pm 0.015}$ | $0.059_{\pm 0.007}$ | $0.084_{\pm 0.018}$ | $0.176_{\pm 0.012}$ | $0.103_{\pm 0.015}$ |
| iTransformer | 0.052M | 50 | $0.080_{\pm 0.010}$ | $0.099_{\pm 0.011}$ | $0.115_{\pm 0.005}$ | $0.221_{\pm 0.007}$ | $0.138_{\pm 0.006}$ |
| iTransformer | 0.052M | 40 | $0.090_{\pm 0.017}$ | $0.070_{\pm 0.012}$ | $0.101_{\pm 0.010}$ | $0.222_{\pm 0.018}$ | $0.128_{\pm 0.008}$ |
| iTransformer | 0.052M | 60 | $0.110_{\pm 0.020}$ | $0.087_{\pm 0.009}$ | $0.103_{\pm 0.009}$ | $0.187_{\pm 0.004}$ | $0.126_{\pm 0.005}$ |
| FEDformer | 0.1M | 30 | $0.117_{\pm 0.023}$ | $0.114_{\pm 0.022}$ | $0.147_{\pm 0.030}$ | $0.331_{\pm 0.014}$ | $0.190_{\pm 0.015}$ |
| FEDformer | 0.1M | 40 | $0.120_{\pm 0.046}$ | $0.106_{\pm 0.022}$ | $0.129_{\pm 0.016}$ | $0.278_{\pm 0.011}$ | $0.168_{\pm 0.013}$ |
| iTransformer | 0.052M | 30 | $0.123_{\pm 0.025}$ | $0.129_{\pm 0.005}$ | $0.167_{\pm 0.007}$ | $0.328_{\pm 0.012}$ | $0.202_{\pm 0.001}$ |
| FEDformer | 0.1M | 20 | $0.130_{\pm 0.040}$ | $0.157_{\pm 0.006}$ | $0.219_{\pm 0.018}$ | $0.483_{\pm 0.016}$ | $0.272_{\pm 0.015}$ |
| iTransformer | 0.052M | 20 | $0.250_{\pm 0.017}$ | $0.248_{\pm 0.012}$ | $0.293_{\pm 0.021}$ | $0.543_{\pm 0.005}$ | $0.354_{\pm 0.007}$ |
| GVAR | 1.3M | 60 | $0.388_{\pm 0.006}$ | $0.289_{\pm 0.003}$ | $0.312_{\pm 0.003}$ | $0.468_{\pm 0.009}$ | $0.366_{\pm 0.003}$ |
| GVAR | 0.9M | 50 | $0.442_{\pm 0.021}$ | $0.345_{\pm 0.022}$ | $0.353_{\pm 0.017}$ | $0.532_{\pm 0.013}$ | $0.419_{\pm 0.019}$ |
| GVAR | 0.6M | 40 | $0.560_{\pm 0.011}$ | $0.418_{\pm 0.006}$ | $0.436_{\pm 0.019}$ | $0.634_{\pm 0.012}$ | $0.510_{\pm 0.011}$ |
| GVAR | 0.3M | 30 | $0.655_{\pm 0.033}$ | $0.505_{\pm 0.003}$ | $0.538_{\pm 0.006}$ | $0.748_{\pm 0.019}$ | $0.616_{\pm 0.010}$ |
| GVAR | 0.2M | 20 | $0.811_{\pm 0.023}$ | $0.642_{\pm 0.021}$ | $0.665_{\pm 0.016}$ | $0.875_{\pm 0.002}$ | $0.750_{\pm 0.013}$ |
| CrGSTA | 0.7M | 30 | $0.927_{\pm 0.006}$ | $0.738_{\pm 0.008}$ | $0.729_{\pm 0.007}$ | $0.899_{\pm 0.015}$ | $0.816_{\pm 0.007}$ |
| CrGSTA | 1.0M | 40 | $0.930_{\pm 0.017}$ | $0.744_{\pm 0.005}$ | $0.678_{\pm 0.006}$ | $0.848_{\pm 0.004}$ | $0.782_{\pm 0.007}$ |
| CrGSTA | 1.3M | 50 | $0.937_{\pm 0.032}$ | $0.699_{\pm 0.013}$ | $0.627_{\pm 0.005}$ | $0.778_{\pm 0.013}$ | $0.734_{\pm 0.007}$ |
| CrGSTA | 1.7M | 60 | $0.940_{\pm 0.010}$ | $0.707_{\pm 0.015}$ | $0.660_{\pm 0.020}$ | $0.797_{\pm 0.006}$ | $0.755_{\pm 0.008}$ |
| CrGSTA | 0.4M | 20 | $0.950_{\pm 0.010}$ | $\underline{0.789_{\pm 0.014}}$ | $\mathbf{0.786_{\pm 0.014}}$ | $\underline{0.948_{\pm 0.007}}$ | $\mathbf{0.866_{\pm 0.004}}$ |
| causalrca | 0.003M | 30 | $0.963_{\pm 0.006}$ | $0.732_{\pm 0.007}$ | $0.662_{\pm 0.007}$ | $0.835_{\pm 0.000}$ | $0.769_{\pm 0.005}$ |
| causalrca | 0.009M | 60 | $0.963_{\pm 0.006}$ | $0.734_{\pm 0.007}$ | $0.628_{\pm 0.006}$ | $0.695_{\pm 0.003}$ | $0.720_{\pm 0.005}$ |
| causalrca | 0.002M | 20 | $0.964_{\pm 0.003}$ | $0.767_{\pm 0.007}$ | $0.728_{\pm 0.005}$ | $\mathbf{0.956_{\pm 0.000}}$ | $0.844_{\pm 0.003}$ |
| causalrca | 0.005M | 40 | $0.965_{\pm 0.003}$ | $0.748_{\pm 0.008}$ | $0.645_{\pm 0.005}$ | $0.766_{\pm 0.002}$ | $0.749_{\pm 0.004}$ |
| AERCA | 0.5M | 20 | $0.965_{\pm 0.007}$ | $\mathbf{0.822_{\pm 0.002}}$ | $\underline{0.772_{\pm 0.004}}$ | $0.926_{\pm 0.008}$ | $\underline{0.859_{\pm 0.003}}$ |
| AERCA | 2.8M | 50 | $0.965_{\pm 0.021}$ | $0.742_{\pm 0.007}$ | $0.650_{\pm 0.019}$ | $0.769_{\pm 0.020}$ | $0.749_{\pm 0.012}$ |
| causalrca | 0.007M | 50 | $0.970_{\pm 0.003}$ | $0.735_{\pm 0.007}$ | $0.619_{\pm 0.002}$ | $0.702_{\pm 0.000}$ | $0.717_{\pm 0.004}$ |
| AERCA | 1.1M | 30 | $0.970_{\pm 0.028}$ | $0.772_{\pm 0.007}$ | $0.727_{\pm 0.018}$ | $0.873_{\pm 0.020}$ | $0.821_{\pm 0.005}$ |
| AERCA | 1.8M | 40 | $\underline{0.985_{\pm 0.007}}$ | $0.760_{\pm 0.038}$ | $0.690_{\pm 0.026}$ | $0.797_{\pm 0.004}$ | $0.784_{\pm 0.016}$ |
| AERCA | 4.0M | 60 | $\mathbf{0.990_{\pm 0.000}}$ | $0.773_{\pm 0.014}$ | $0.691_{\pm 0.008}$ | $0.794_{\pm 0.006}$ | $0.788_{\pm 0.004}$ |

Table 12: RQ2 -Lotka Voltera - Spatial Scaling

A.9.2.2   NonLinear Dataset Results.

| scheme | Params | num vars | AC@1 | AC@3 | AC@5 | AC@10 | Avg@10 |
|---|---|---|---|---|---|---|---|
| NONLINEAR | | | | | | | |
| FEDformer | 0.1M | 35 | $0.140_{\pm0.017}$ | $0.157_{\pm0.008}$ | $0.180_{\pm0.004}$ | $0.274_{\pm0.012}$ | $0.190_{\pm0.001}$ |
| FEDformer | 0.1M | 30 | $0.147_{\pm0.023}$ | $0.196_{\pm0.001}$ | $0.228_{\pm0.017}$ | $0.355_{\pm0.003}$ | $0.239_{\pm0.007}$ |
| iTransformer | 0.051M | 35 | $0.147_{\pm0.006}$ | $0.176_{\pm0.002}$ | $0.192_{\pm0.001}$ | $0.278_{\pm0.002}$ | $0.203_{\pm0.001}$ |
| GVAR | 0.4M | 35 | $0.177_{\pm0.055}$ | $0.189_{\pm0.055}$ | $0.199_{\pm0.029}$ | $0.308_{\pm0.036}$ | $0.224_{\pm0.038}$ |
| FEDformer | 0.1M | 25 | $0.180_{\pm0.010}$ | $0.224_{\pm0.012}$ | $0.245_{\pm0.007}$ | $0.379_{\pm0.007}$ | $0.267_{\pm0.007}$ |
| CrGSTA | 0.8M | 35 | $0.197_{\pm0.031}$ | $0.223_{\pm0.012}$ | $0.234_{\pm0.019}$ | $0.370_{\pm0.015}$ | $0.264_{\pm0.008}$ |
| AERCA | 1.2M | 35 | $0.197_{\pm0.049}$ | $0.186_{\pm0.015}$ | $0.189_{\pm0.020}$ | $0.307_{\pm0.037}$ | $0.220_{\pm0.024}$ |
| GVAR | 0.3M | 30 | $0.213_{\pm0.032}$ | $0.223_{\pm0.033}$ | $0.250_{\pm0.014}$ | $0.394_{\pm0.005}$ | $0.282_{\pm0.017}$ |
| iTransformer | 0.051M | 30 | $0.223_{\pm0.006}$ | $0.207_{\pm0.007}$ | $0.253_{\pm0.001}$ | $0.381_{\pm0.001}$ | $0.271_{\pm0.001}$ |
| causalrca | 0.004M | 35 | $0.227_{\pm0.021}$ | $0.226_{\pm0.013}$ | $0.259_{\pm0.009}$ | $0.370_{\pm0.002}$ | $0.277_{\pm0.006}$ |
| AERCA | 1.0M | 30 | $0.237_{\pm0.042}$ | $0.239_{\pm0.020}$ | $0.274_{\pm0.026}$ | $0.385_{\pm0.006}$ | $0.293_{\pm0.019}$ |
| iTransformer | 0.051M | 25 | $0.240_{\pm0.000}$ | $0.202_{\pm0.002}$ | $0.206_{\pm0.001}$ | $0.363_{\pm0.003}$ | $0.254_{\pm0.001}$ |
| FEDformer | 0.1M | 20 | $0.273_{\pm0.006}$ | $0.295_{\pm0.009}$ | $0.341_{\pm0.008}$ | $0.535_{\pm0.009}$ | $0.375_{\pm0.002}$ |
| iTransformer | 0.051M | 20 | $0.290_{\pm0.010}$ | $0.316_{\pm0.004}$ | $0.329_{\pm0.003}$ | $0.524_{\pm0.004}$ | $0.374_{\pm0.001}$ |
| AERCA | 0.8M | 25 | $0.290_{\pm0.046}$ | $0.283_{\pm0.013}$ | $0.312_{\pm0.020}$ | $0.435_{\pm0.029}$ | $0.334_{\pm0.016}$ |
| CrGSTA | 0.7M | 30 | $0.293_{\pm0.059}$ | $0.307_{\pm0.015}$ | $0.325_{\pm0.014}$ | $0.464_{\pm0.010}$ | $0.354_{\pm0.011}$ |
| RCD | 0.000M | 15 | $0.310_{\pm0.000}$ | $0.377_{\pm0.000}$ | $0.436_{\pm0.000}$ | $0.677_{\pm0.000}$ | $0.477_{\pm0.000}$ |
| GVAR | 0.3M | 25 | $0.313_{\pm0.032}$ | $0.287_{\pm0.007}$ | $0.323_{\pm0.014}$ | $0.455_{\pm0.034}$ | $0.348_{\pm0.022}$ |
| AERCA | 0.6M | 20 | $0.363_{\pm0.023}$ | $0.339_{\pm0.053}$ | $0.377_{\pm0.055}$ | $0.555_{\pm0.058}$ | $0.412_{\pm0.049}$ |
| GVAR | 0.2M | 20 | $0.368_{\pm0.038}$ | $0.366_{\pm0.041}$ | $0.383_{\pm0.033}$ | $0.537_{\pm0.032}$ | $0.417_{\pm0.033}$ |
| CrGSTA | 0.5M | 25 | $0.397_{\pm0.035}$ | $0.362_{\pm0.036}$ | $0.381_{\pm0.021}$ | $0.521_{\pm0.034}$ | $0.417_{\pm0.022}$ |
| iTransformer | 0.051M | 15 | $0.406_{\pm0.016}$ | $0.399_{\pm0.010}$ | $0.446_{\pm0.010}$ | $0.693_{\pm0.006}$ | $0.498_{\pm0.004}$ |
| causalrca | 0.003M | 30 | $0.427_{\pm0.012}$ | $0.346_{\pm0.010}$ | $0.344_{\pm0.020}$ | $0.453_{\pm0.010}$ | $0.384_{\pm0.012}$ |
| FEDformer | 0.1M | 15 | $0.430_{\pm0.013}$ | $0.420_{\pm0.009}$ | $0.456_{\pm0.011}$ | $0.693_{\pm0.009}$ | $0.502_{\pm0.004}$ |
| Epsilon | 0.000M | 15 | $0.440_{\pm0.000}$ | $0.445_{\pm0.000}$ | $0.452_{\pm0.000}$ | $0.687_{\pm0.000}$ | $0.512_{\pm0.000}$ |
| causalrca | 0.003M | 25 | $0.493_{\pm0.012}$ | $0.441_{\pm0.018}$ | $0.452_{\pm0.015}$ | $0.570_{\pm0.006}$ | $0.489_{\pm0.012}$ |
| causalrca | 0.002M | 20 | $0.500_{\pm0.031}$ | $0.477_{\pm0.028}$ | $0.474_{\pm0.027}$ | $0.608_{\pm0.021}$ | $0.515_{\pm0.025}$ |
| CrGSTA | 0.4M | 20 | $0.503_{\pm0.045}$ | $0.481_{\pm0.028}$ | $0.492_{\pm0.018}$ | $0.641_{\pm0.022}$ | $0.521_{\pm0.024}$ |
| AERCA | 0.5M | 15 | $0.503_{\pm0.074}$ | $0.500_{\pm0.039}$ | $0.537_{\pm0.041}$ | $0.744_{\pm0.024}$ | $0.579_{\pm0.037}$ |
| GVAR | 0.2M | 15 | $0.520_{\pm0.046}$ | $0.505_{\pm0.059}$ | $\underline{0.568}_{\pm0.042}$ | $0.743_{\pm0.026}$ | $0.597_{\pm0.038}$ |
| CrGSTA | 0.3M | 15 | $\underline{0.608}_{\pm0.037}$ | $\underline{0.566}_{\pm0.013}$ | $\mathbf{0.591}_{\pm0.012}$ | $\mathbf{0.791}_{\pm0.014}$ | $\mathbf{0.637}_{\pm0.008}$ |
| causalrca | 0.002M | 15 | $\mathbf{0.610}_{\pm0.000}$ | $\mathbf{0.567}_{\pm0.007}$ | $0.567_{\pm0.007}$ | $\underline{0.759}_{\pm0.010}$ | $\underline{0.621}_{\pm0.003}$ |

Table 13: RQ2 - NonLinear - Spatial Scaling

To assess spatial scalability on the NonLinear dataset, we fix the temporal window (5) and vary the number of variables (Table 13).

**Causal vs. Non-Causal Models.** As in Lotka–Volterra, causal models consistently outperform non-causal baselines. FEDformer and iTransformer show modest gains at small scales but degrade quickly as dimensionality increases (e.g., at 35 variables both remain below 0.28 Avg@10). In contrast, causal methods maintain substantially higher accuracy: causalrca reaches 0.489–0.515 Avg@10 at 20–25 variables and remains competitive even at 35 variables. This highlights that structural modeling is essential for capturing the stronger nonlinear interactions in this dataset. **CrGSTA Performance.** CrGSTA demonstrates consistently strong and stable performance across all dimensionalities. At 35 variables it reaches 0.264 Avg@10—outperforming all non-causal baselines—and scales robustly down to 20 variables, where it achieves 0.521. CrGSTA also performs best at lower dimensions: at 15 variables, it achieves the highest overall Avg@10 (0.637), outperforming GVAR (0.597), AERCA (0.579), and causalrca (0.621). These results confirm that CrGSTA handles both moderate and small-scale nonlinear dynamics effectively. **Parameter Efficiency.** Across all scales, CrGSTA remains highly parameter-efficient. For example, at 20 variables it achieves 0.521 Avg@10 with only 0.4M parameters—substantially smaller than AERCA (0.412 with 0.6M). At 30–35 vari-

ables, CrGSTA maintains competitive performance with 0.7–0.8M parameters, outperforming larger causal models such as AERCA (1.0–1.2M) and sharply exceeding the accuracy of non-causal baselines with similar or larger parameter counts.

**Summary.** On the NonLinear dataset, CrGSTA delivers state-of-the-art spatial scalability, outperforming non-causal models at all dimensionalities while remaining competitive with or superior to larger causal baselines. Its strong accuracy at both low and high dimensions—paired with its compact parameter footprint—demonstrates its suitability for nonlinear, high-dimensional dynamical systems.

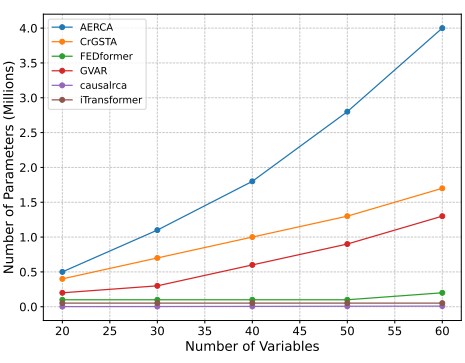 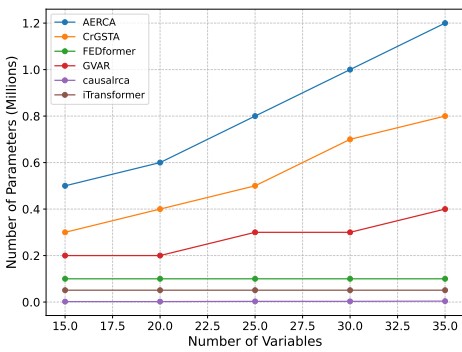

(a) Lotka Volterra Parameters

(b) Non-Linear Parameters

Figure 9: Parameter scaling for Spatial dimension Lotka Volterra (left) and Non-Linear (right) for temporal scaling.

### A.9.3 RQ3 (ABLATIONS) - FULL TABLES

#### A.9.3.1 Baselines and Ablations

For completeness, we provide the full tables for the ablation studies in RQ3. Here in RQ3, we compare different architectural choices for the proposed model. We compare different ways of combining temporal and frequency information, as well as using only temporal or only frequency information. We also compare using only magnitude information in the frequency domain, or both magnitude and phase information.

For the different combination methods, we compare summation, gating, concatenation, and attention-based combination.

**Sum**: Element-wise summation of the two representations, as shown in Eq. 50, where $H_T$ is the temporal representation and $H_F$ is the frequency representation.

$$H = H_T + H_F \tag{50}$$

**Concat**: Concatenation of the two representations followed by a linear layer to reduce the dimension back to the original, as shown in Eq. 51.

$$H = W \cdot [H_T; H_F] + b \tag{51}$$

where $W$ and $b$ are learnable parameters. Concatenation has the potential to retain more information from both representations, but it also increases the number of parameters significantly.

**Gated**: A gating mechanism to control the contribution of each representation, as shown in Eq. 52.

$$g = \sigma(W_g \cdot [H_T; H_F] + b_g) H = g * H_T + (1 - g) * H_F \tag{52}$$

where $W_g$ and $b_g$ are learnable parameters, and $\sigma$ is the sigmoid function. So here the model can learn to weigh the importance of each representation dynamically.

**Attention**: Cross attention mechanism where one representation attends to the other, here it is composed of two cross-attention modules, as shown in Eq. 53.

$$\tilde{\mathbf{H}}^{\text{time}} = \text{CrossAttn}(\mathbf{H}_t, \mathbf{H}^{\text{freq}}), \quad \tilde{\mathbf{H}}^{\text{freq}} = \text{CrossAttn}(\mathbf{H}^{\text{freq}}, \mathbf{H}_t). \tag{53}$$

Which are then combined as seen in step 5 of CrGSTA in the main paper.

As shown in Tables 14 and 15, we observe several clear trends:

**Domains.** Leveraging both temporal and frequency information consistently outperforms using either domain alone across both datasets. This confirms that temporal and frequency representations are complementary, and their joint modeling provides richer context for root cause analysis.

**Cross Attention.** Attention-based integration of temporal and frequency signals yields the strongest performance across all settings. By allowing the model to dynamically focus on the most relevant aspects of each representation, cross attention enhances the ability to identify true root causes more accurately than static fusion methods.

**Parameter Efficiency.** Figure 10 reports parameter counts for each configuration. Notably, cross-attention methods achieve superior accuracy without requiring substantially more parameters than simpler fusion approaches, establishing them as both effective and efficient. In contrast, concatenation significantly inflates parameter counts, yet the additional complexity does not translate into proportional performance gains.

**Phase Information.** Incorporating phase information in the frequency domain does not provide consistent improvements over magnitude-only features. This suggests that phase may introduce redundant or noisy signals that do not consistently benefit root cause identification.

**Summary.** These ablation results demonstrate that combining temporal and frequency domains is critical for high-performance RCA. Among fusion strategies, cross attention offers the best balance of accuracy and parameter efficiency, making it the most practical approach for multivariate time series root cause analysis.

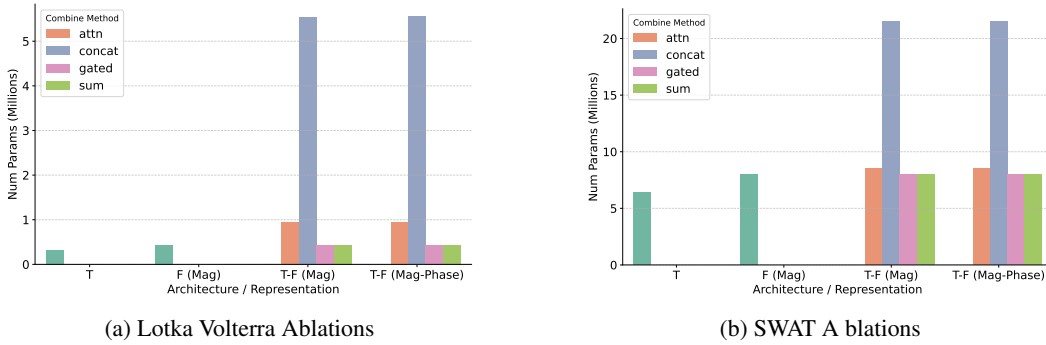

(a) Lotka Volterra Ablations        (b) SWAT A blations

Figure 10: Parameters for ablations for Lotka Volterra (left) and SWAT (right).

| Model | Temp | Freq | Mag | Phase | Fusion Type | Params | AC@1 | AC@3 | Avg@10 |
|---|---|---|---|---|---|---|---|---|---|
| Lotka Volterra | | | | | | | | | |
| Freq Only (Mag) | ✗ | ✓ | ✓ | ✗ | – | 0.4M | 0.730 | 0.434 | 0.482 |
| T–F (Mag–Phase, concat) | ✓ | ✓ | ✓ | ✓ | concat | 5.6M | 0.708 | 0.459 | 0.517 |
| T–F (Mag, concat) | ✓ | ✓ | ✓ | ✗ | concat | 5.5M | 0.722 | 0.463 | 0.522 |
| T–F (Mag, sum) | ✓ | ✓ | ✓ | ✗ | sum | 0.4M | 0.720 | 0.462 | 0.525 |
| T–F (Mag–Phase, sum) | ✓ | ✓ | ✓ | ✓ | sum | 0.4M | 0.720 | 0.468 | 0.526 |
| Temporal Only | ✓ | ✗ | – | ✗ | – | 0.3M | 0.767 | 0.487 | 0.546 |
| T–F (Mag–Phase, gated) | ✓ | ✓ | ✓ | ✓ | gated | 0.4M | 0.787 | 0.525 | 0.571 |
| T–F (Mag, gated) | ✓ | ✓ | ✓ | ✗ | gated | 0.4M | 0.790 | 0.529 | 0.575 |
| T–F (Mag, attn) | ✓ | ✓ | ✓ | ✗ | attn | 0.9M | **0.893** | 0.603 | 0.639 |
| T–F (Mag–Phase, attn) | ✓ | ✓ | ✓ | ✓ | attn | 1.0M | 0.893 | **0.604** | **0.639** |

Table 14: Ablation results on Lotka-Voltera. Best in bold, second best underlined.

| Model | Temp | Freq | Mag | Phase | Fusion Type | Params | AC@1 | AC@3 | Avg@10 |
|---|---|---|---|---|---|---|---|---|---|
| | | | | SWaT | | | | | |
| Temporal Only | ✓ | ✗ | – | ✗ | – | 6.4M | 0.213 | 0.297 | 0.334 |
| T–F (Mag–Phase, concat) | ✓ | ✓ | ✓ | ✓ | concat | 21.6M | 0.210 | 0.299 | 0.340 |
| T–F (Mag, gated) | ✓ | ✓ | ✓ | ✗ | gated | 8.0M | 0.213 | 0.299 | 0.344 |
| T–F (Mag, sum) | ✓ | ✓ | ✓ | ✗ | sum | 8.0M | 0.201 | 0.318 | 0.351 |
| T–F (Mag–Phase, sum) | ✓ | ✓ | ✓ | ✓ | sum | 8.0M | 0.179 | 0.295 | 0.352 |
| T–F (Mag, concat) | ✓ | ✓ | ✓ | ✗ | concat | 21.5M | 0.198 | 0.311 | 0.355 |
| T–F (Mag–Phase, gated) | ✓ | ✓ | ✓ | ✓ | gated | 8.0M | 0.258 | 0.327 | 0.360 |
| Freq Only (Mag) | ✗ | ✓ | ✓ | ✗ | – | 8.0M | 0.242 | 0.320 | 0.365 |
| T–F (Mag–Phase, attn) | ✓ | ✓ | ✓ | ✓ | attn | 8.5M | **0.312** | **0.396** | 0.425 |
| T–F (Mag, attn) | ✓ | ✓ | ✓ | ✗ | attn | 8.5M | 0.311 | 0.395 | **0.430** |

Table 15: Ablation results on SWAT. Best in bold, second best underlined.

### A.9.4 RQ 4 - CASE STUDY DETAILS

In this section, we provide the detailed data used in the case studies for both the msds using AERCA and CrGSTA models.

Table 16: Case-study data for MSDS AERCA.

| Variable | Fused z-score | Root Cause |
|---|---|---|
| 0 | 0.318 | ✗ |
| 1 | -1.100 | ✗ |
| 2 | -4.251 | ✗ |
| 3 | 6.595 | ✗ |
| 4 | -1.188 | ✗ |
| 5 | 1.985 | ✗ |
| 6 | -1.352 | ✗ |
| 7 | 6.455 | ✗ |
| 8 | 0.406 | ✗ |
| 9 | 2.990 | ✓ |

Table 17: Case-study data for MSDS CrGSTA.

| Variable | Fused z-score | Root Cause |
|---|---|---|
| 0 | -0.806 | ✗ |
| 1 | -4.150 | ✗ |
| 2 | -4.325 | ✗ |
| 3 | 1.769 | ✗ |
| 4 | -0.403 | ✗ |
| 5 | 1.181 | ✗ |
| 6 | -1.429 | ✗ |
| 7 | 3.004 | ✗ |
| 8 | 0.099 | ✗ |
| 9 | 2.604 | ✓ |

## B USE OF LLMs

We used GPT-5 from ChatGPT and Copilot to help with writing and refining the text and code in this paper.

