# OpenReview forum: "CrGSTA: Cross-domain Root causal Graph Spatial-Temporal Attention Network"
_ICLR.cc/2026/Conference — Submitted to ICLR 2026_

### Official Review · Reviewer_mxam · 2025-10-30

**Soundness:** 3
**Presentation:** 2
**Contribution:** 3
**Rating:** 4
**Confidence:** 3

**Summary:**

The paper tackles root cause analysis (RCA) for high-dimensional multivariate time series, arguing that existing neural Granger approaches miss long-range dependencies, seasonal/periodic structure, and scale poorly. It proposes CrGSTA—a Cross-domain Root causal Graph Spatio-Temporal Attention network. CrGSTA unifies time- and frequency-domain views with a shared spatial graph attention backbone, temporal attention on lags/bins, and bi-directional cross-attention to fuse domains. The encoder yields interpretable, Granger-style coefficients and latent exogenous variables; a lightweight self-attention decoder reconstructs dynamics. Emperical experiments demonstrate that CrGSTA is effective and helps improve performance compared with other models.

Contributions:
1) An interpretable, parameter-efficient RCA framework that couples spatial graph attention with time–frequency modeling.
2) A multi-path encoder–decoder with cross-domain fusion that captures periodicity/seasonality while preserving causal interpretability.
3) Extensive experiments shows SOTA performance.

**Strengths:**

1) The cross-attention mechanism between time and frequency domains is novel for RCA tasks, and targeted important and practical problem of this domain.
2) Results and analysis indicate practical value for large-scale industrial monitoring where both interpretability and compute budget matter. Also the model design directly target the theoretical deficit of previous works, which helps explain the performance improvement with experiment demonstration.
3) Detailed reproduce informations and settings.

**Weaknesses:**

1) The discussion on related work for root cause analysis can be improved to be more thorough and coherent. In particular, incorporating discussion and comparison with more works utilizing causal graphs, such as Wang, D., Chen, Z., Fu, Y., Liu, Y., & Chen, H. (2023, August). Incremental causal graph learning for online root cause analysis. In Proceedings of the 29th ACM SIGKDD Conference on Knowledge Discovery and Data Mining (pp. 2269–2278), will help improve the quality.
2) The space utilization in the main plot could be improved. For example, the title text appears relatively small while noticeable blank areas remain unused. Adding descriptive captions would also help readers follow the figures more easily. In addition, the presentation of the experimental section could be more clearly structured, with explicit motivation, experimental settings, result interpretation, and concluding remarks.
3) The evaluation currently only compares against AERCA among causal methods. Including additional recent RCA approaches would provide a more comprehensive and convincing comparison.
4) The clarity of the ablation table may be enhanced by clearly highlighting the ablated components, which would help readers better interpret the contribution of each module.
5) Table-1 in Appendix is somehow misleading, “Comparison of Root Cause Analysis (RCA) and Anomaly Detection Approaches”, since there are large portion of RCA and Anomaly detection approaches, the table content does not justify the title and intention well.

**Questions:**

1) Could you discuss additional causal graph–based RCA methods and compare them with this paper? Including these works in the related-work section would further improve the coherence and thoroughness of the analysis.
2) If adaptable, incorporating comparisons with more causal models in the experimental section would further substantiate the effectiveness claims of the proposed method.
3) Improving space utilization and adding a clearer caption for Figure 1 may enhance readability. For the ablation study plots, explicitly tagging the ablated components would also improve clarity.
4) The material presented in “Table 1: Comparison of Root Cause Analysis (RCA) and Anomaly Detection Approaches” reads more like a feature comparison than a conceptual justification of the methods. It would be helpful to justify the table name or adjust it to ensure that the title and the presented content are aligned.

---

> ### Author Response · Authors · 2025-11-20
> **(revised related works), (improved fig and model explanations), (2 new baselines), (2 new datasets), (2 new case studies)**
>
> Thanks for your valuable feedback. We address your points as follows:
>
> W1/Q1: We revised the related work section to include additional recent causal graph–based RCA methods, including the suggested work by Wang et al. (2023). We also added GVAR, another recent causal graph–based RCA method that AERCA builds upon. These additions provide a more coherent and thorough context for our proposed method within the landscape of existing approaches.
>
> W2/Q3: We improved space utilization in Figure 1 by providing two versions: a simplified version in the main text to highlight the parallel-then-fusion structure, and a detailed architecture diagram in the appendix. All equations now have descriptive explanations in the appendix, and we included motivations for each architectural component to clarify their roles and contributions (A.3, A.4, A.5, A.6). Moreover, we have included a motivation for the parameter configuration choices across the different baselines (A.8.1)
>
> W3/Q2: To strengthen the experimental comparison, we added two new causal baselines (GVAR and CausalRCA), providing a more comprehensive benchmarking against recent RCA approaches. We included these baselines in both RQ1 and RQ2, to better highlight the effectiveness of CrGSTA in both the temporal and spatial (num of variables) dimensions. Moreover, we have added 2 new datasets, and added 2 new case studies in RQ4.
>
> W4/Q3: The ablation table has been revised to clearly indicate ablated components using a checkmark system, allowing readers to easily interpret each module’s contribution to overall performance.
>
> W5/Q4: Table 1 in the appendix has been updated to better align the title with its content. The new title, “Related Works Comparison of Generic Time Series Models and Root Cause Analysis Methods,” accurately reflects the scope and intent of the table.
>
>
> In summary, we have addressed the reviewers’ feedback as follows:
>
>     Sec. 2 (Related Work):
>         - Expanded discussion of causal graph–based RCA methods (e.g., CORAL, GVAR).
>
>     Sec. 3 (Model):
>         - Simplified Figure 1 in the main text; detailed architecture moved to the appendix.
>         - Added detailed explanations of all equations in the appendix.
>         - Provided justifications for each architectural component.
>
>     Sec. 4 (Experiments):
>         - Added two new datasets (MSDS, Non-linear).
>         - Added two new causal baselines (GVAR, CausalRCA), reflected in RQ1 and RQ2.
>         - Introduced an additional RQ4 as a case study on interpretability of learned causal graphs.
>
>     Appendix:
>         - Revised Table 1 title for clarity.
>         - Related works section has been strengthened for coherence and completeness.
>         - Revised ablation table to clearly highlight ablated components.

---

### Official Review · Reviewer_E8VB · 2025-10-30

**Soundness:** 3
**Presentation:** 3
**Contribution:** 2
**Rating:** 4
**Confidence:** 4

**Summary:**

This paper presents CrGSTA (Cross-domain Root causal Graph Spatial-Temporal Attention Network), a framework for interpretable root cause analysis (RCA) in multivariate time series. CrGSTA integrates temporal and frequency-domain representations via cross-domain attention and employs a graph-based encoder-decoder to capture both spatial and temporal dependencies. The method builds upon Granger causality for interpretability, uses graph attention networks to model variable relationships, and applies self-attention for efficient temporal modeling. Extensive experiments on the Lotka–Volterra synthetic dataset and the SWaT industrial dataset demonstrate that CrGSTA achieves competitive or superior results to statistical, non-causal, and causal baselines (e.g., AERCA), with fewer parameters and improved interpretability.

**Strengths:**

+ Well-motivated integration of temporal, frequency, and causal reasoning.
+ Strong empirical performance on both synthetic and real-world benchmarks.
+ Comprehensive ablation studies validating design choices.

**Weaknesses:**

- This paper is primarily an integration of existing mechanisms (attention, GNN, Granger causality).
- The evaluation scope is limit. It only consider two datasets, both within standard RCA benchmark. There exist many public multivariate time series (MTS) benchmarks such as SMD, SMAP, MSL.Testing on these would better demonstrate the generalization ability.
- Some dense mathematical sections could benefit from summarizing intuition alongside formulas.
- The paper could show more qualitative evidence (e.g., visualized causal graphs or case studies) to confirm interpretability in practice.

**Questions:**

1. How well does the model generalize across domains (e.g., trained on SWaT, tested on other industrial datasets)?
2. Does the frequency-domain branch improve interpretability, performance, or both?
3. What is the computational complexity compared with AERCA for long sequences?

**Details Of Ethics Concerns:**

The submitted paper includes a GitHub link (https://github.com/crgsta2025/CrGSTA).
Upon inspection, the repository’s LICENSE file explicitly states “Copyright (c) 2025 Xiao Han,” revealing an author’s name.
This appears to violate the ICLR double-blind policy by directly exposing author identity.
I did not further investigate or attempt to confirm the identity.
Please advise whether this should be treated as a serious anonymization breach.

---

> ### Author Response · Authors · 2025-11-20
> **(model better explanation), (2 new datasets, 2 new baselines), (2 new case studies), (freq-domain branch highlighting its benefits)**
>
> We appreciate the reviewer’s thoughtful and constructive comments. Below we address each point in turn.
>
> W1: Thank you for the feedback. While CrGSTA integrates existing mechanisms such as attention, GNNs, and Granger causality, its novelty lies in how these components are combined to address multivariate time series root cause analysis. Specifically, CrGSTA jointly optimizes causal interpretability and parameter efficiency, a balance not achieved by prior methods such as AERCA and GVAR, which either emphasize interpretability at the cost of extremely high parameter counts or are inefficient for long sequences.
>
> W2: For this paper, we have added two new datasets in the revised version: a real-world cloud dataset (MSDS) and a nonlinear synthetic dataset. We have increased the complexity of the nonlinear dataset compared to the one used in AERCA, similar to our approach with Lotka–Volterra, to better reflect real-world scenarios. Additionally, we include two new causal baselines, GVAR and CausalRCA, to broaden the empirical evaluation. The details of our modifications for both synthetic datasets are provided in the appendix.
>
> W3: To improve readability, we added intuitive explanations alongside all mathematical formulations in the appendix. Figure 1 has been simplified to highlight the parallel-then-fusion structure, while the detailed architecture is now in the appendix. Each architectural component is explicitly justified, clarifying its role and motivation.
>
> W4: We added a new RQ4 as a qualitative case study on interpretability. This includes visualization of learned causal graphs, z-score trajectories for a selected window, and temporal stability across windows. These analyses confirm that CrGSTA produces interpretable and robust causal structures across diverse domains.
>
> Questions
>
> Q1 (cross-domain generalization):
> We thank the reviewer for the suggestion. In this work, we focus primarily on how CrGSTA’s architecture captures the intrinsic dependencies within a given dataset. Exploring explicit cross-domain or transfer learning is indeed an interesting avenue for future work. For now, we have included an additional real-world dataset (cloud monitoring) alongside SWaT, showing that CrGSTA maintains robust performance across systems with differing characteristics, highlighting its ability to capture short- and long-term dependencies, oscillatory dynamics, and latent causal structures.
>
> Q2 (frequency-domain branch):
> The frequency-domain branch contributes to both interpretability and performance. For real-world datasets, even before time–frequency fusion, the Frequency-only model (F (Mag)) outperforms the Time-only model (T), as shown in Fig. 4(b), emphasizing the importance of frequency-domain analysis for capturing periodic or oscillatory dependencies.  Moreover, the time-freq attention improves performance (Avg@10) while using far fewer parameters than alternative fusion methods, such as concatenation, as shown in Fig. 10(b).
> For the additional MSDS dataset, the time–frequency cross-attention further enhances interpretability compared to AERCA by highlighting the most relevant spectral components for root-cause analysis, as demonstrated in RQ4.
>
> Q3 (computational complexity vs. AERCA):
> AERCA, similar to GVAR, repeats separate MLPs for each lag, leading to a large parameter footprint and high computational cost, especially for long sequences. We have shown the number of parameters in Fig. 8 in the appendix for RQ1.
> CrGSTA instead uses a shared architecture with attention mechanisms to model temporal dependencies, dramatically reducing both parameter count and computational complexity. This design enables efficient training and inference for long sequences, making CrGSTA scalable compared to AERCA.
>
> Ethics concern:
> We apologize for the oversight regarding the LICENSE file in the GitHub repository. The repository was forked from AERCA’s codebase, and we forgot to remove the LICENSE. The file reflects the original AERCA authorship not our own.
> https://github.com/hanxiao0607/AERCA/blob/main/LICENSE
>
> In summary, we have addressed the reviewers’ feedback as follows:
>
> Sec. 2 (Related Work):
> - Expanded discussion of causal graph–based RCA methods (e.g., CORAL, GVAR).
>
> Sec. 3 (Model):
> - Simplified Figure 1 in the main text; detailed architecture moved to the appendix.
> - Added detailed explanations of all equations in the appendix.
> - Provided justifications for each architectural component.
>
> Sec. 4 (Experiments):
> - Added two new datasets (MSDS, Non-linear).
> - Added two new causal baselines (GVAR, CausalRCA), reflected in RQ1 and RQ2.
> - Introduced an additional RQ4 as a case study on interpretability of learned causal graphs.
>
> Appendix:
> - Revised Table 1 title for clarity.
> - Related works section has been strengthened for coherence and completeness.
> - Revised ablation table to clearly highlight ablated components.

---

> > ### Comment · Reviewer_E8VB · 2025-11-25
> > **maintain my score**
> >
> > Thanks for the authors' response, but I still have **doubts regarding the novelty** of the work, and I think the current contribution is incremental. On another note, I would suggest the authors highlight some key points in the rebuttal and try to include specific results. Therefore, I **still maintain my scores**.

---

> > > ### Author Response · Authors · 2025-11-26
> > > **(for MSDS Higher accuracy with 0.7% params , for SWAT comparable accuracy with 12% params), (efficient spatial scaling), (cross-domain time-freq attn), (interpretable results), (more datasets, more baselines)**
> > >
> > > Thank you for the additional feedback.
> > > Below, we summarize the key innovations and provide concrete empirical evidence from the updated manuscript.
> > > We also added two new datasets (MSDS, nonlinear synthetic), two new causal baselines (GVAR, CausalRCA), and two new interpretability case studies (z-score visualization and temporal graph stability).
> > >
> > > ---
> > >
> > > ### **1. Temporal Scaling**
> > >
> > > CrGSTA achieves comparable or better accuracy while using only **12%** of the parameters of GVAR and **4%** of the parameters for AERCA on SWaT (w=7):
> > >
> > > summary of table 9, fig2.b, fig8.b for SWaT (w=7)
> > > | Model      | Avg@10    | Params (M) |
> > > | ---------- | --------- | ---------- |
> > > | GVAR       | 0.462     | 67.6       |
> > > | AERCA      | —         | 202.9      |
> > > | **CrGSTA** | **0.426** | **8.5**    |
> > >
> > > CrGSTA incurs only a **4% accuracy drop** relative to GVAR while using **12%** of its parameters. AERCA cannot run due to OOM/time constraints, while it can theoretically achieve higher results, it would have done this incurring more than 200M params, while CrGSTA only uses 8.5M (CrGSTA uses **4%** of AERCA params)
> > >
> > > On MSDS (w=3), CrGSTA is dramatically more efficient:
> > >
> > > summary of table 11, fig 8b, fig 2d
> > > | Model      | Avg@10    | Params (M) |
> > > | ---------- | --------- | ---------- |
> > > | GVAR       | 0.859     | 9.3        |
> > > | AERCA      | 0.864     | 28.0       |
> > > | **CrGSTA** | **0.937** | **0.069**  |
> > >
> > > CrGSTA achieves the best accuracy with only **0.7%** of GVAR’s parameters and **0.25%** of AERCA’s.
> > > This is due to that using attention dimension of size 16 in the encoder was enough in CrGSTA to capture the temporal dependencies for RCA in MSDS
> > > while for GVAR and AERCA, as per AERCA's paper, need larger MLPs to capture the temporal dependencies, and still cannot achieve the same accuracy as CrGSTA
> > >
> > > ---
> > >
> > > ### **2. Spatial Scaling**
> > >
> > > CrGSTA maintains the best accuracy–parameter tradeoff as the number of variables increases, thanks to adapter-based input layers that grow slowly while keeping the attention backbone fixed.
> > >
> > > summary of table 12, fig 9a, fig 3a for Lotka-Volterra
> > >
> > > **#Vars = 20**
> > >
> > > | Model      | Avg@10    | Params (M) |
> > > | ---------- | --------- | ---------- |
> > > | GVAR       | 0.750     | 0.2        |
> > > | AERCA      | 0.859     | 0.5        |
> > > | **CrGSTA** | **0.866** | **0.4**    |
> > >
> > > **#Vars = 60**
> > >
> > > | Model      | Avg@10    | Params (M) |
> > > | ---------- | --------- | ---------- |
> > > | GVAR       | 0.366     | 1.3        |
> > > | AERCA      | 0.788     | 4.0        |
> > > | **CrGSTA** | **0.755** | **1.7**    |
> > >
> > > GVAR uses fewer parameters but collapses in accuracy; AERCA scales nearly linearly in parameters (0.5→4.0M). CrGSTA remains the best balance.
> > >
> > > ---
> > >
> > > ### **3. Cross-Domain Time–Frequency Attention**
> > >
> > > Frequency-domain features outperform time-only baselines on complex systems like SWaT:
> > > summary of table 15, fig4.b for SWaT
> > >
> > > | Method                  | Avg@10 |
> > > | ----------------------- | ------ |
> > > | Time only (T)           | 0.334  |
> > > | Frequency only (F, Mag) | 0.365  |
> > >
> > > Cross-domain attention significantly outperforms naive fusion and uses fewer parameters than concatenation:
> > >
> > > | Fusion Method     | Avg@10    | Params (M) |
> > > | ----------------- | --------- | ---------- |
> > > | T+F (sum)         | 0.351     | 8.0        |
> > > | gated             | 0.344     | 8.0        |
> > > | concatenation     | 0.355     | 21.5       |
> > > | **CrGSTA (ours)** | **0.430** | **8.5**    |
> > >
> > > ---
> > >
> > > ### **4. Interpretability Case Studies**
> > >
> > > We add two new interpretability analyses:
> > >
> > > * **z-score anomaly visualization**, showing how CrGSTA localizes faults
> > > * **temporal stability of the causal graphs**, demonstrating robust causal reasoning over time
> > >
> > > ---
> > >
> > > ### **Summary**
> > >
> > > Across **temporal scaling**, **spatial scaling**, **cross-domain attention**, and **interpretability**, CrGSTA introduces several innovations. The expanded experiments and additional baselines consistently demonstrate that CrGSTA achieves new Pareto frontiers in **accuracy, efficiency, and causal interpretability**.

---

### Official Review · Reviewer_WnrN · 2025-10-31

**Soundness:** 2
**Presentation:** 2
**Contribution:** 2
**Rating:** 2
**Confidence:** 3

**Summary:**

This paper introduces CrGSTA, a scalable and interpretable deep learning framework for root cause analysis in multivariate time series. CrGSTA addresses the parameter inefficiency and limited pattern-capturing capacity of prior MLP-based approaches by adopting a multi-path encoder–decoder architecture. The model jointly leverages time-domain and frequency-domain representations through cross-attention and incorporates graph-based spatio-temporal attention to capture complex inter-variable relationships. Experimental results on the Lotka–Volterra and SWaT benchmarks demonstrate that CrGSTA achieves state-of-the-art performance while maintaining significantly higher parameter efficiency, highlighting its practicality for large-scale, high-dimensional monitoring scenarios.

**Strengths:**

- The paper is clearly written and easy to follow, with well-structured explanations that support reader comprehension.
-  The proposed multi-path encoder—integrating parallel time- and frequency-domain representations with cross-domain attention fusion—is well-motivated and technically sound.
-  The model achieves strong empirical performance, matching or surpassing state-of-the-art baselines while using up to 20× fewer parameters, demonstrating notable efficiency and a well-validated architectural design through thorough ablation studies.

**Weaknesses:**

- The experimental evaluation is limited to only two datasets (Lotka–Volterra and SWaT). Incorporating additional benchmarks—as done in AERCA [1]—would help strengthen the generalizability and external validity of the findings.
-  Although the framework is positioned as interpretable, no human-in-the-loop evaluation or rule-based validation is provided. A qualitative case study that links attention weights or learned coefficients to known process topology would substantially reinforce the interpretability claim.
-  The magnitude–phase component appears to yield only marginal gains, as it does not consistently improve performance across the ablation studies, raising questions about its practical benefit to the overall architecture.

[1] Han, X., Absar, S., Zhang, L., & Yuan, S. (2025). Root Cause Analysis of Anomalies in Multivariate Time Series through Granger Causal Discovery. The Thirteenth International Conference on Learning Representations.
- The main contribution and claim of the paper are not clearly stated. The introduction initially suggests a focus on causality, but the technical sections and experiments emphasize parameter efficiency. It remains unclear how the proposed method supports or interprets causality. Providing a qualitative analysis—such as illustrating how the model identifies or explains causal relationships—would strengthen this claim.

-  The writing and methodological explanations require further elaboration.
1.	Several architectural components are introduced without sufficient justification. The paper should explain why each module is needed, how they differ from components in existing models, and how the equations reflect these design intentions. For instance, how are the weights and QKV mappings computed, and what motivates the bidirectional interaction between temporal (time → frequency) and spectral (frequency → time) contexts?
2.	The evaluation is limited in dataset diversity. Prior work such as AERCA evaluates on six datasets (Linear, Nonlinear, Lotka–Volterra, Lorenz-96, SWaT, MSDS), and similar breadth would strengthen claims of generalizability.
3.	The experimental comparison set could be expanded. Additional RCA-based or GNN-based approaches should be included to provide a more comprehensive benchmarking against current methods.

**Questions:**

1. In Section 4.3, the authors state that “Results are presented in Fig. 2 and summarized in Tables 8 and 9.” However, the content in these tables does not correspond to Figure 2 Did the authors intend to refer to Tables 5 and 6 instead?
2.  In the SWaT dataset results, the reported performance of AERCA is shown as 0 for window sizes 5, 7, 10, and 12. Could the authors clarify whether this indicates a failure mode, an implementation issue, or missing results?
3.  In the Sparsity & Smoothness loss formulation, the symbol Ωis introduced but not defined. What does Ωrepresent in this context?
4.  On the Lotka–Volterra benchmark, CrGSTA’s performance shows large variation as the window size increases, while other baseline models remain comparatively stable. Could the authors provide an explanation for this sensitivity?
4. AERCA seems to be higher performance in many places, e.g., Figure 2(a) 0.791 and Table 5
5. Why there are 0.000s in the result (e.g., Table 6). This may indicate the evaluation is not fair.

---

> ### Author Response · Authors · 2025-11-20
> **(2 new datasets, 2 new baselines), (2 new case studies), (zeros clarification), (model better explanation),**
>
> We appreciate the reviewer’s thoughtful and constructive comments. Below we address each point in turn.
>
> W1: We have expanded our evaluation to include two additional datasets: a real-world cloud dataset (MSDS) and a more challenging nonlinear synthetic dataset. We also incorporated two additional causal baselines (GVAR and CausalRCA) to further strengthen the empirical scope and generalizability of our results.
>
> W2: We have added a new RQ4 as a qualitative case study on interpretability. This includes visualization of learned causal graphs, z-score trajectories for a selected window, and temporal stability analyses across windows. These results demonstrate that CrGSTA produces interpretable structures aligned with known system dynamics.
>
> W3: We thank the reviewer for their careful reading and for noticing this point. We acknowledge that the magnitude–phase component yields only modest improvements and does not consistently enhance performance across our ablations. Nonetheless, we have included it for completeness, as it may provide value to researchers working with more challenging real-time datasets exhibiting oscillatory or synchronization-driven behavior, where phase information could carry meaningful causal cues. We also acknowledge the importance of this avenue, which we found essential for further investigation in future work.
>
> W4: We have clarified that CrGSTA advances both causal interpretability and parameter efficiency. It inherits structural causal interpretability through Granger-style modeling while avoiding the parameter blow-up of prior methods like AERCA and GVAR. The bidirectional time–frequency interaction allows the model to identify causal drivers across temporal scales, which purely time-domain models struggle with. To make this explicit, we added RQ4 as a case study on the interpretability of the learned causal graphs, including z-scores for one window and their temporal stability across windows. The appendix now provides detailed explanations of each equation and the motivation behind architectural components to improve clarity on how the model identifies causal relationships.
>
> W5: We added detailed explanations of all architectural components and equations to the appendix and revised the main text for clarity. We also expanded the comparison (GVAR and CausalRCA), and increased dataset diversity with MSDS and a more complex nonlinear synthetic dataset, aligning the breadth of our evaluation with prior work such as AERCA.
>
> Q1: Thank you for pointing out the inconsistency. We have corrected the reference to point to the appropriate appendix section, as the revised version includes additional tables for the newly added datasets.
>
> Q2/Q6:
> We apologize for not making this explicit earlier, we have fixed both the fig and the tables
> As shown in Fig. 2(b), we adopt a 100M-parameter threshold for any baseline. AERCA exceeds this threshold for all window sizes greater than 1 due to its lag-specific MLP replication, which results in severe training-time inflation for small windows and out-of-memory errors for extreme windows (e.g., 12). We have clarified this explicitly in both the figure and table captions in the revised version.
>
> Q3: Thank you for noting this. We now explicitly define $\omega$ in the Sparsity \& Smoothness loss section.
>
> Q4:
> The larger variation of CrGSTA across window sizes on Lotka–Volterra arises from the dataset’s highly nonlinear, oscillatory dynamics, where causal influences alternate between short- and long-range dependencies. Models without cross-domain (time and frequency) processing tend to smooth over these transitions, yielding more stable but less accurate performance. CrGSTA’s time-frequency fusion is sensitive to temporal context: small windows may miss longer predator-prey cycles, while large windows capture periodicity but also regime-switching segments. This sensitivity is beneficial because CrGSTA achieves higher peak accuracy than all baselines when the window aligns with the dataset’s intrinsic periodicity. On other datasets (Nonlinear, MSDS, SWaT), this behavior is smoother, indicating that the variation reflects the dataset’s dynamics rather than model instability.
>
> Q5: We thank the reviewer for pointing this out. The higher performance of AERCA (e.g., Figure 2(a) 0.791, Table 5) is mainly observed on synthetic datasets with limited nonlinearity. AERCA and GVAR follow their original architectures, allowing parameter counts to expand with window size, so the improved performance largely reflects increased parameters rather than architectural advantage. In contrast, CrGSTA fixes the attention dimension relative to the temporal dimension, deliberately limiting parameters to stress-test the architecture itself. Despite this “unfair” comparison, CrGSTA achieves higher Avg@10 values with far fewer parameters, demonstrating that its strength arises from architectural design, not from parameter growth, making it both accurate and practically efficient.

---

> > ### Comment · Reviewer_WnrN · 2025-11-25
> >
> > Thanks for the response. I raised my score.

---

### Official Review · Reviewer_wZAR · 2025-11-01

**Soundness:** 2
**Presentation:** 3
**Contribution:** 3
**Rating:** 4
**Confidence:** 3

**Summary:**

This paper presents CrGSTA, a novel deep learning framework for Root Cause Analysis (RCA) in high-dimensional, complex multivariate time series (MTS). The work is motivated by a key limitation in existing neural Granger causality methods (e.g., AERCA), which often rely on per-time-step MLPs. The authors argue this approach scales poorly, increasing parameters dramatically with longer temporal windows, and fails to capture periodic or seasonal patterns critical for RCA.

To solve this, CrGSTA proposes a unified encoder-decoder architecture grounded in Granger causality (where the "root cause" is the unexplained exogenous residual, $z_t$). Experiments on a synthetic (Lotka-Volterra) and a real-world (SWaT) dataset show that CrGSTA achieves new state-of-the-art results (e.g., +13% Avg@10 on Lotka-Volterra).

**Strengths:**

S1. The paper addresses a critical and practical problem: scalable, interpretable RCA. It correctly identifies the trade-off between accuracy and parameter/computational cost in prior SOTA causal models as a major bottleneck.

S2. The experimental results and their in-depth analysis are insightful.

S3. The results for RQ2 (Fig 3) demonstrate that CrGSTA also scales more efficiently with the number of variables (spatial dimension), using significantly fewer parameters (1.7M) than AERCA (4.0M) to achieve comparable performance at 60 variables.

**Weaknesses:**

W1. One of the concerns is the reported performance of the primary causal baseline, AERCA, on the SWaT dataset. In Table 6 and Figure 2b, AERCA scores 0.000 on all metrics (AC@1, Avg@10, etc.) for all window sizes. This is highly ridiculous. It implies the model failed completely (worse than random chance). Did it fail to run, suffer from OOM, or was there an implementation bug? This is a critical comparison, and the 0.000 result is not adequately explained, undermining the perceived gap between CrGSTA and the prior SOTA.

W2. The paper states AERCA's parameter count "grows to over 100M" and "200M+" (Sec 4.3). Figure 5b shows it reaching 350M. This is a key selling point, but the implementation details for AERCA in Table 4 mention "8 layers (1000 nodes) per lag." This seems to be the authors' own implementation choice, and it's unclear if this is a necessary or fair configuration for the AERCA baseline. A brief justification for why AERCA was configured to be so large is needed.

W3. The model is exceptionally complex. While the ablation justifies this, it makes the paper very dense and challenging to parse. Figure 1, in particular, is a difficult "boxes and arrows" diagram that obscures the elegance of the parallel-then-fused design.

W4. The paper's evaluation is deep but narrow. All claims are supported by strong results on only two datasets: one synthetic (Lotka-Volterra) and one real-world (SWaT). While these are good benchmarks, the strong claims of the paper would be far more generalizable if validated on at least one other real-world domain (e.g., cloud microservice, finance, or a different CPS dataset).

**Questions:**

Q1.  The 0.000 Avg@10 score for AERCA on SWaT (Table 6, Fig 2b) is a major concern. Please clarify this result. Did the baseline implementation fail to run, OOM, or did it genuinely produce zero correct results? This is the most critical point to address.

Q2. The model's validation is strong but limited to Lotka-Volterra and SWaT. How do you expect CrGSTA's cross-domain (Time+Freq) approach to perform on systems that are less governed by physical laws or clear periodicity, such as a stochastic cloud microservice environment?

Q3. Could you elaborate on the parameter configuration for the AERCA baseline? Is the "8 layers (1000 nodes) per lag" setting (Table 4) derived from the original AERCA paper, or was this a choice made for this paper?

Q4. The ablation (Fig 4) shows that for SWaT, the Frequency-only model (F (Mag)) outperforms the Time-only model (T) and even naive T+F fusion. This is a very interesting finding. Does this imply that for some real-world CPS systems, frequency-domain analysis is more important than time-domain for RCA?

---

> ### Author Response · Authors · 2025-11-20
> **(zeros bec of OOM or prohibitively long training time), (params from AERCA), (more datasets and more baselines), (2 new case studies)**
>
> We appreciate the reviewer’s thoughtful feedback and constructive comments. Below are our detailed responses to each point.
>
> W1 / Q1:
> Sorry for this confusion, we have clarified this.
> As is now shown in Fig. 2b, we adopt a strict threshold of 100M parameters for all baselines. Models exceeding this limit are considered impractical, as they are more than 10× larger than CrGSTA, often cause OOM failures for large windows, or require prohibitively long training time. AERCA crossed this threshold on SWaT for most window sizes,
> while GVAR also exceeded it for window of size 12. We have clarified this policy directly in Fig. 2b and updated the corresponding tables to avoid ambiguity. We apologize for not making this explicit earlier.
>
> W2 / Q3:
> Although AERCA reaches 300M+ parameters for larger windows, we conservatively report the value (~200M) corresponding to the best-performing CrGSTA window (window 7) to ensure a fair and honest comparison.
> Regarding the parameter configuration of AERCA, we strictly follow the official implementation for SWaT (8 layers of 1000 units per lag), as defined in the authors’ code repository https://github.com/hanxiao0607/AERCA/blob/main/args/swat_args.py#L37.
> This configuration is indeed highly parameter-inefficient because AERCA, like GVAR, replicates a full MLP stack for each lag, causing parameter count to scale linearly with window size. This inefficiency was a primary motivation for developing CrGSTA.
> Moreover, we have added a detailed motivation for the parameter configuration in the appendix in the revised version to clarify this point.
>
> W3:
> We appreciate the comment regarding architectural complexity. In response, we have simplified Figure 1 to emphasize the parallel-then-fusion structure while relocating the full detailed architecture diagram to the appendix. We also expanded the intuitive explanation accompanying the mathematical formulations and provided clearer justification for each architectural component to improve readability and conceptual accessibility in the appendix (A.2, A.3, A.4, A.5, A.6).
>
> W4 / Q2:
> We agree that broader evaluation strengthens the generality of our claims. In the revised version, we include two additional datasets: a real-world cloud microservice dataset (MSDS) and a more challenging synthetic nonlinear dataset.
> The inclusion of MSDS demonstrates CrGSTA’s effectiveness in stochastic cloud environments, which are less governed by physical laws or periodicity compared to CPS systems. The nonlinear synthetic dataset further tests the model’s robustness against complex dynamics.
> We also add two new causal baselines (GVAR and CausalRCA). Furthermore, we introduce a new RQ4 presenting a case study on interpretability, including z-score visualizations across windows and a temporal-stability analysis of inferred causal graphs. These additions demonstrate CrGSTA’s robustness across both physical and stochastic environments and reinforce the generalizability of our cross-domain (time + frequency) approach.
>
> Q4:
> Thank you for highlighting this insight. We now discuss in Section 4.5 that, for certain CPS settings such as SWaT, frequency-domain evidence can dominate time-domain cues due to periodic actuation cycles, noise filtering effects, and sensor co-movement patterns. This observation provides an interesting avenue for future work on adaptive domain weighting.
>
>
> In summary, we have addressed the reviewers’ feedback as follows:
>
> Sec. 2 (Related Work):
> - Expanded discussion of causal graph–based RCA methods (e.g., CORAL, GVAR).
>
> Sec. 3 (Model):
> - Simplified Figure 1 in the main text; detailed architecture moved to the appendix.
> - Added detailed explanations of all equations in the appendix.
> - Provided justifications for each architectural component.
>
> Sec. 4 (Experiments):
> - Added two new datasets (MSDS, Non-linear), reflected in RQ1 and RQ2.
> - Added two new causal baselines (GVAR, CausalRCA), reflected in RQ1 and RQ2.
> - Introduced an additional RQ4 as a case study on interpretability of learned causal graphs.
> - Added clarifications regarding parameter configuration across the different baselines.
>
> Appendix:
> - Revised Table 1 title for clarity.
> - Related works section has been strengthened for coherence and completeness.
> - Revised ablation table to clearly highlight ablated components.

---

### Author Response · Authors · 2025-11-26
**(for MSDS Higher accuracy with 0.7% params , for SWAT comparable accuracy with 12% params), (efficient spatial scaling), (cross-domain time-freq attn), (interpretable results), (more datasets, more baselines)**

We thank the reviewers for their careful reading and constructive feedback. Here, we summarize the most important points that were raised by multiple reviewers and highlight the key contributions and novelties of our work.

In particular, we emphasize:
- higher accuracy on MSDS with only **0.7%** of the parameters, comparable accuracy on SWaT using just **12%** of the parameters,
- efficient spatial scaling,
- cross-domain time–frequency attention,
- interpretable results, and
- the addition of new datasets and causal baselines.

To strengthen the manuscript, we have added two new datasets (MSDS and nonlinear synthetic), two new causal baselines (GVAR and CausalRCA), and two interpretability case studies (z-score visualization and temporal z-score stability).
Moreover, all equations now have descriptive explanations in the appendix, and we included motivations for each architectural component to clarify their roles and contributions.
These additions demonstrate the efficiency, accuracy, and practical relevance of CrGSTA across multiple benchmarks and provide further evidence of its novelty and generalizability.

Below is a summary of these additions.

---

### **1. Temporal Scaling**

CrGSTA achieves comparable or better accuracy while using only **12%** of the parameters of GVAR and **4%** of the parameters for AERCA on SWaT (w=7):

summary of table 9, fig2.b, fig8.b for SWaT (w=7)
| Model      | Avg@10    | Params (M) |
| ---------- | --------- | ---------- |
| GVAR       | 0.462     | 67.6       |
| AERCA      | —         | 202.9      |
| **CrGSTA** | **0.426** | **8.5**    |

CrGSTA incurs only a **4% accuracy drop** relative to GVAR while using **12%** of its parameters. AERCA cannot run due to OOM/time constraints, while it can theoretically achieve higher results, it would have done this incurring more than 200M params, while CrGSTA only uses 8.5M (CrGSTA uses **4%** of AERCA params)

On MSDS (w=3), CrGSTA is dramatically more efficient:

summary of table 11, fig 8b, fig 2d
| Model      | Avg@10    | Params (M) |
| ---------- | --------- | ---------- |
| GVAR       | 0.859     | 9.3        |
| AERCA      | 0.864     | 28.0       |
| **CrGSTA** | **0.937** | **0.069**  |

CrGSTA achieves the best accuracy with only **0.7%** of GVAR’s parameters and **0.25%** of AERCA’s.
This is due to that using attention dimension of size 16 in the encoder was enough in CrGSTA to capture the temporal dependencies for RCA in MSDS
while for GVAR and AERCA, as per AERCA's paper, need larger MLPs to capture the temporal dependencies, and still cannot achieve the same accuracy as CrGSTA

---

### **2. Spatial Scaling**

CrGSTA maintains the best accuracy–parameter tradeoff as the number of variables increases, thanks to adapter-based input layers that grow slowly while keeping the attention backbone fixed.

summary of table 12, fig 9a, fig 3a for Lotka-Volterra

**#Vars = 20**

| Model      | Avg@10    | Params (M) |
| ---------- | --------- | ---------- |
| GVAR       | 0.750     | 0.2        |
| AERCA      | 0.859     | 0.5        |
| **CrGSTA** | **0.866** | **0.4**    |

**#Vars = 60**

| Model      | Avg@10    | Params (M) |
| ---------- | --------- | ---------- |
| GVAR       | 0.366     | 1.3        |
| AERCA      | 0.788     | 4.0        |
| **CrGSTA** | **0.755** | **1.7**    |

GVAR uses fewer parameters but collapses in accuracy; AERCA scales nearly linearly in parameters (0.5→4.0M). CrGSTA remains the best balance.

---

### **3. Cross-Domain Time–Frequency Attention**

Frequency-domain features outperform time-only baselines on complex systems like SWaT:
summary of table 15, fig4.b for SWaT

| Method                  | Avg@10 |
| ----------------------- | ------ |
| Time only (T)           | 0.334  |
| Frequency only (F, Mag) | 0.365  |

Cross-domain attention significantly outperforms naive fusion and uses fewer parameters than concatenation:

| Fusion Method     | Avg@10    | Params (M) |
| ----------------- | --------- | ---------- |
| T+F (sum)         | 0.351     | 8.0        |
| gated             | 0.344     | 8.0        |
| concatenation     | 0.355     | 21.5       |
| **CrGSTA (ours)** | **0.430** | **8.5**    |

---

### **4. Interpretability Case Studies**

We add two new interpretability analyses:

* **z-score anomaly visualization**, showing how CrGSTA localizes faults
* **temporal stability of the z-score**, demonstrating robust causal reasoning over time

---

### **Summary**

Across **temporal scaling**, **spatial scaling**, **cross-domain attention**, and **interpretability**, CrGSTA introduces several innovations. The expanded experiments and additional baselines consistently demonstrate that CrGSTA achieves new Pareto frontiers in **accuracy, efficiency, and causal interpretability**.

---

### Meta-Review · Area_Chair_zpeD · 2026-01-05

**Summary:**

The paper introduces CrGSTA, a novel framework designed for root cause analysis (RCA) in high-dimensional multivariate time series data. It seeks to address limitations in current neural Granger causality methods, which often struggle with scaling and capturing complex patterns. CrGSTA integrates time- and frequency-domain representations through cross-domain attention, enabling it to effectively model causal relationships. The model demonstrates state-of-the-art performance with significant parameter efficiency, achieving improved accuracy while using far fewer parameters than existing models.

Novelty and Incremental Contribution: Some reviewers expressed concerns regarding the novelty of CrGSTA, suggesting that the contributions may be seen as incremental rather than groundbreaking.

Limited Experimental Scope: The evaluation was primarily based on two datasets (Lotka-Volterra and SWaT), raising questions about generalizability outside these benchmarks. Reviewers noted the need for broader testing against additional datasets.

Parameter Efficiency Claims: Although CrGSTA claims to have significantly lower parameter counts, reviewers raised concerns about the configurations used in baseline comparisons, particularly with the AERCA model.

Interpretability Claims: The interpretability of the model was questioned, with suggestions for qualitative analysis to substantiate claims regarding the model's ability to map causal relationships.

**Reviewer Concerns:**

Addressing Novelty: The authors emphasized CrGSTA's unique integration of components (such as cross-domain attention) and its focus on balancing interpretability with parameter efficiency. This was acknowledged by Reviewer WnrN who raised their score. However, Reviewer E8VB still had doubts about the novelty.

Expanding Experimental Validation: In response to concerns about the limited scope, the authors added two new datasets and two causal baselines, enhancing the manuscript's empirical breadth and reinforcing the generalizability of their findings. This was acknowledged by Reviewer WnrN who raised their score.

Clarifying Parameter Comparisons: The authors provided additional context regarding the parameter configurations used for AERCA and highlighted the efficiency of their approach through improved explanations.

Strengthening Interpretability: The manuscript now includes additional case studies and interpretability analyses to illustrate how CrGSTA identifies and explains causal relationships, supporting its claims on interpretability.

**Reviewer Scores:**

As mentioned previously, Reviewer WnrN seemed to be convinced by the authors' responses and raised their score from 2. However, it is not visible to me what the Reviewer WnrN raised their score to. Even if it's a 6, the resultant average score would still be too low for acceptance.

Reviewer E8VB still had doubts about the novelty of the work. Having read the paper, I tend to take the view of Reviewer E8VB that the integration of the various components does not pass the bar for publication at ICLR.

Given the reviewers' mixed assessments, particularly regarding the novelty and potential limitations in the breadth of the experimental evaluation, it seems prudent for me to lean towards a weak rejection for this ICLR submission. While the authors satisfactorily addressed many concerns, lingering doubts about the significance of the contributions and the potential for anonymity breach indicate that the paper may not yet satisfy the conference's acceptance criteria. I encourage the authors to exercise caution in their future submissions.

Lastly, please refrain from altering the default font of the paper, as it complicates readability and is unfair to other authors who adhere to the standard template.

---

### Decision · Program_Chairs · 2026-01-26

Reject